# Theoretically Guaranteed Bidirectional Data Rectification for Robust Sequential Recommendation

**Yatong Sun**[1,2], **Xiaochun Yang**[1*], **Zhu Sun**[3*], **Bin Wang**[1], **Yan Wang**[2]

[1]School of Computer Science and Engineering, Northeastern University, China
[2]School of Computing, Macquarie University, Australia
[3]Center for Frontier AI Research, Institute of High Performance Computing, A*STAR, Singapore
yatong@stumail.neu.edu.cn, yangxc@mail.neu.edu.cn, sunzhuntu@gmail.com,
binwang@mail.neu.edu.cn, yan.wang@mq.edu.au

## Abstract

Sequential recommender systems (SRSs) are typically trained to predict the next item as the *target* given its preceding (and succeeding) items as the *input*. Such a paradigm assumes that every input-target pair is reliable for training. However, users can be induced to click on items that are inconsistent with their true preferences, resulting in unreliable instances, i.e., mismatched input-target pairs. Current studies on mitigating this issue suffer from two limitations: (i) they discriminate instance reliability according to models trained with unreliable data, yet without theoretical guarantees that such a seemingly contradictory solution can be effective; and (ii) most methods can only tackle either unreliable input or targets but fail to handle both simultaneously. To fill the gap, we theoretically unveil the relationship between SRS predictions and instance reliability, whereby two error-bounded strategies are proposed to rectify unreliable targets and input, respectively. On this basis, we devise a model-agnostic **Bi**di**r**ectional **D**ata **Rec**tification (**BirDRec**) framework, which can be flexibly implemented with most existing SRSs for robust training against unreliable data. Additionally, a rectification sampling strategy is devised and a self-ensemble mechanism is adopted to reduce the (time and space) complexity of BirDRec. Extensive experiments on four real-world datasets verify the generality, effectiveness, and efficiency of our proposed BirDRec.

## 1 Introduction

Recently, the study on sequential recommender systems (SRSs) [1, 2, 3, 4, 5] has garnered much attention as users' preferences are inherently dynamic and evolving in real-world scenarios. The goal of SRSs is learning to predict the next item a user interacts with given the preceding (and succeeding) items. Therefore, a training instance for SRSs is typically composed of an *input* item sequence and its next item as the *target*. However, distractions in daily lives (e.g. recommendations from friends, account sharing, and accidental clicks) can induce users to click on items that are inconsistent with their true preferences, resulting in unreliable training instances with mismatched input-target pairs. The mismatch can be categorized into *Complete Mismatch* and *Partial Mismatch* when the item caused by distractions acts as an unreliable target and unreliable input of an instance, respectively. To illustrate, the romantic film 'La La Land', in the first instance of Figure 1, serves as an unreliable target, which is recommended by friends and completely mismatched with the previous superhero movies. By contrast, in the second instance, 'La La Land' acts as an unreliable input item which renders the input sequence partially mismatched with the target superhero film. Both

---

*denotes the corresponding authors

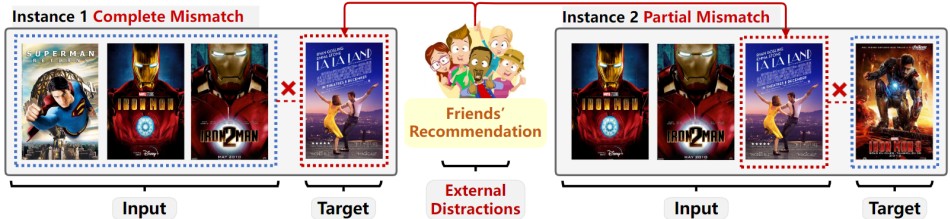

Figure 1: Examples of two types of mismatch caused by external distractions.

types of mismatch would cause unreliable training instances that mislead SRSs to learn sequential relationships between irrelevant items and eventually undermine the recommendation accuracy.

Although there are a number of studies aiming to combat such unreliable data for more robust SRSs, they suffer from two core limitations. (i) They discriminate instance reliability based on the intermediate or final output of a model (either the SRS itself [6, 7, 8, 9] or an additional corrector [10]) that is trained with unreliable data. However, there is no theoretical guarantee that such seemingly contradictory solutions can be trustworthy for detecting and correcting unreliable instances. (ii) Most prior studies only focus on tackling either unreliable input [6, 7, 11, 12, 13] or targets [8], but fail to handle both simultaneously. Only one recently proposed method [10] attempts to address this issue, but it relies on a corrector trained with unreliable data, yet without any theoretical guarantees.

As such, we, *for the first time*, theoretically unveil the relationship between the SRS predictions and instance reliability, proving that a target with consistently low prediction scores is unlikely to be reliable, assuming that the randomness of user behavior is limited. It then inspires us to devise two error-bounded rectification strategies to (1) detect consistently low-scored targets (i.e., unreliable targets) and replace them with steadily high-scored items and (2) detect and delete consistently low-scored items within the input (i.e., unreliable input), where the score is estimated by a backward SRS. Note that the unreliable input items, as the interruptions in the input, are not replaced but directly removed to bridge the preceding and succeeding items. Based on these strategies, we propose a model-agnostic **Bi**directional **D**ata **Rec**tification (**BirDRec**) framework which contains two SRSs in opposite directions for tackling both unreliable targets and input. In addition, to reduce the time complexity, a rectification sampling strategy is devised to efficiently obtain consistently high-scored items; to reduce space complexity, a self-ensemble mechanism [14] is adopted to approximate the weighted average prediction scores across different training epochs.

**Contributions. (1)** We are the first to provide theoretically guaranteed data rectification strategies based on SRS predictions to tackle both unreliable input and targets for more robust SRSs. **(2)** We devise a model-agnostic bidirectional data rectification framework that can be flexibly implemented with most existing SRSs for robust training against unreliable data. **(3)** We devise a rectification sampling strategy and adopt a self-ensemble mechanism to ensure better scalability of BirDRec. **(4)** Extensive experiments with SRSs based on representative backbones and datasets across various domains validate the generality, effectiveness, and efficiency of BirDRec.

## 2 Related Works

Early SRSs [15, 16] adopt Markov Chains to encode users' interaction sequences, assuming users' latest interactions affect the future behavior linearly and independently. Later, powerful deep learning backbones such as recurrent neural networks (RNNs) [17, 18, 19], convolution neural networks (CNNs) [20, 21, 22], graph neural networks (GNNs) [23, 24, 25, 26, 27], and Transformers [28, 29] are employed to extract complex non-linear patterns within users' sequences [30]. They posit each training instance is a definitely matched input-target pair and thus cannot handle unreliable data.

To resist unreliable data, existing robust SRSs can be categorized into three types. The *first type* focuses on handling the complete mismatch by identifying and eliminating instances with unreliable targets. For example, BERD [8] empirically finds that instances with high training loss and low uncertainty tend to have unreliable targets. This idea is relevant to studies on clean label selection [31, 32, 33, 34, 35] and label correction [36, 37, 38]. The *second type* concentrates on addressing partial mismatch by reducing the importance of unreliable input when formulating users' dynamic preference representations. Accordingly, various advanced mechanisms are integrated into SRSs, such as memory networks [7], gating networks [39, 40, 41], autoencoders [12], reinforcement learning [6],

uncertainty modelling [11, 42], and Fast Fourier Transform [9]. To the best of our knowledge, there is only one recently proposed method STEAM [10] falling into the *third type* which attempts to tackle both unreliable targets and input with an additional corrector producing reliable data. Nonetheless, existing robust SRSs all rely on models (either SRSs or additional correctors) trained with unreliable data, yet without theoretical proof that such seemingly contradictory solutions can be effective.

## 3 Theoretical Guarantees for Rectifying Unreliable Data

This section presents theoretical guarantees for rectifying unreliable data via SRS predictions for more robust SRSs. In particular, Section 3.1 introduces important preliminaries. Subsequently, Section 3.2 unveils the relationship between the prediction score of SRSs and the reliability of a target, inspiring us to propose an error-bounded strategy for handling unreliable targets; and Section 3.3 provides the error-bounded strategy for dealing with unreliable input.

### 3.1 Preliminaries

**Problem Statement of SRSs**. Let $\mathcal{U}$ and $\mathcal{V}$ be the sets of users and items, respectively. Each user $u \in \mathcal{U}$ chronologically interacts with a sequence of items $\mathbf{s}^u = [\tilde{v}_1^u, \tilde{v}_2^u, ..., \tilde{v}_{|\mathbf{s}^u|}^u]$, where $\tilde{v}_t^u \in \mathcal{V}$ is the $t$-th item user $u$ interacts with and $|\mathbf{s}^u|$ is the length of sequence $\mathbf{s}^u$. The goal of SRSs is to predict the target item $\tilde{v}_t^u$ given the input $\tilde{\mathbf{x}}_t^u = \left\{ u, [\tilde{v}_{t-L}^u, ..., \tilde{v}_{t-2}^u, \tilde{v}_{t-1}^u] \right\}$, where $L$ is the length of $\tilde{\mathbf{x}}_t^u$. Thus, the training instance of SRSs can be represented as an input-target pair $\langle \tilde{\mathbf{x}}_t^u, \tilde{v}_t^u \rangle$. Note that we use '~' to denote the observed data that may be unreliable due to external distractions.

**Core Assumptions**. Ideally, each user-item interaction should be drawn from users' true preference distribution $\eta$ without any distractions, where $\eta_{v_i}(\mathbf{x}_t^u) = \mathbb{P}(v_t^u = v_i | \mathbf{x}_t^u)$. We define $p_1$ to be the true item for recommendation, i.e, the top-1 item according to $\eta$,

$$\sum\nolimits_{v_i \in \mathcal{V}} \mathbb{I}\big[\eta_{p_1}(\mathbf{x}_t^u) \geq \eta_{v_i}(\mathbf{x}_t^u)\big] = |\mathcal{V}|, \tag{1}$$

where $\mathbb{I}[\cdot]$ is an indicator function that equals 1 if the condition is true; otherwise 0. Meanwhile, we define $p_2$ to be the middle-ranked item (ranked $\lfloor |\mathcal{V}|/2 \rfloor$-th) according to $\eta$, namely,

$$\sum\nolimits_{v_i \in \mathcal{V}} \mathbb{I}\big[\eta_{p_2}(\mathbf{x}_t^u) \geq \eta_{v_i}(\mathbf{x}_t^u)\big] = \lfloor |\mathcal{V}|/2 \rfloor. \tag{2}$$

In general, SRSs are built upon the hypothesis that users usually select items with a tendency rather than randomly. In other words, the randomness of users' true preferences is restricted, i.e., the probability gap between the top-1 and middle-ranked items regarding $\eta$ is unlikely to be small. This assumption can be formally defined as follows.

**Assumption 1.** *The users' true preference distribution $\eta$ fulfills the relaxed Multiclass Tsybakov Condition [43] with constants $C > 0$, $\lambda > 0$, and $\alpha_0 \in (0, 1]$, such that for all $\alpha \in (0, \alpha_0]$,*

$$\mathbb{P}\big[\eta_{p_1}(\mathbf{x}_t^u) - \eta_{p_2}(\mathbf{x}_t^u) \leq \alpha\big] \leq C\alpha^{\lambda}. \tag{3}$$

The feasibility of Assumption 1 relies on small $C$ and large $\lambda$, which are satisfied on public (observed) datasets based on our empirical analysis in the Appendix with $C \in (0.55, 0.70)$ and $\lambda \in (1.37, 4.01)$.

**Connecting $\eta$ with SRS Predictions**. Obviously, $\eta$ is the ideal corrector to rectify unreliable data, however, due to its unavailability, many existing methods [8, 10] leverage SRS predictions as the substitution with no theoretical guarantees. This urges us to explore the connection between $\eta$ and SRS predictions. To achieve this goal, we first investigate the relationship between $\eta$ and users' observed preference distribution $\tilde{\eta}$, since SRSs are trained with the observed data that may be distorted by external distractions. Formally, $\tilde{\eta}_{v_i}(\tilde{\mathbf{x}}_t^u) = \mathbb{P}(\tilde{v}_t^u = v_i | \tilde{\mathbf{x}}_t^u)$, where $\tilde{v}_t^u$ is the observed target that may be unreliable. We then define a transition probability $\tau_{v_j v_i}(\tilde{\mathbf{x}}_t^u) = \mathbb{P}(\tilde{v}_t^u = v_i | v_t^u = v_j, \tilde{\mathbf{x}}_t^u)$ as the chance that a true target $v_t^u$ is flipped from item $v_j$ to item $v_i$ owing to external distractions. Thus, for any pair $(v_i, v_j) \in \mathcal{V}$, there is a linear relationship between $\eta$ and $\tilde{\eta}$:

$$\tilde{\eta}_{v_i}(\tilde{\mathbf{x}}_t^u) = \sum\nolimits_{v_j \in \mathcal{V}} \mathbb{P}(\tilde{v}_t^u = v_i | v_t^u = v_j, \tilde{\mathbf{x}}_t^u)\mathbb{P}(v_t^u = v_j | \tilde{\mathbf{x}}_t^u) = \sum\nolimits_{v_j \in \mathcal{V}} \tau_{v_j v_i}(\tilde{\mathbf{x}}_t^u)\eta_{v_j}(\tilde{\mathbf{x}}_t^u). \tag{4}$$

To bridge $\eta$ and SRS predictions via $\tilde{\eta}$, we then study the relationship between $\tilde{\eta}$ and SRS predictions. Let $f^h$ be an SRS at the $h$-th training epoch, and the prediction of $f^h$ be $\epsilon$-close to $\tilde{\eta}$,

$$\epsilon = \max_{\tilde{\mathbf{x}}_t^u, v_i} \big| \tilde{\eta}_{v_i}(\tilde{\mathbf{x}}_t^u) - f_{v_i}^h(\tilde{\mathbf{x}}_t^u) \big|, \tag{5}$$

where $f_{v_i}^h(\tilde{\mathbf{x}}_t^u)$ is the predicted probability (score) of the target being $v_i$ given the input $\tilde{\mathbf{x}}_t^u$ at the $h$-th training epoch. Eqs. (4-5) indicate that there is indeed a connection between $\eta$ and SRS predictions, laying the foundation for the proposed strategies to rectify unreliable data as what follows.

### 3.2 Theorems for Rectifying Unreliable Targets

We now explore how to properly use SRS predictions for rectifying unreliable targets. Prior works empirically find that unreliable targets tend to possess consistently low prediction scores at different epochs [8, 44]. Yet, there is no guarantee that such prediction scores given by an SRS trained with unreliable data can be trustworthy for detecting unreliable targets. This prompts us to theoretically unveil the relationship between target reliability and SRS predictions at different epochs. Specifically, Theorem 1 proves that a reliable target is unlikely to keep low prediction scores during training.

**Theorem 1.** *Given Assumption 1, let* $\{w_h \mid 1 \le h \le H, 0 \le w_h \le 1, \sum_{h=1}^H w_h = 1\}$ *be the weights[1] for averaging prediction scores of different epochs.* $\forall \langle \tilde{\mathbf{x}}_t^u, \tilde{v}_t^u \rangle$, *assume* $\epsilon \le \alpha_0 \tau_{\tilde{v}_t^u \tilde{v}_t^u}(\tilde{\mathbf{x}}_t^u)$. *Let* $\gamma = \tau_{\tilde{v}_t^u \tilde{v}_t^u}(\tilde{\mathbf{x}}_t^u)\eta_{p_2}(\tilde{\mathbf{x}}_t^u) + \sum_{v_j \ne \tilde{v}_t^u} \tau_{v_j \tilde{v}_t^u}(\tilde{\mathbf{x}}_t^u)\eta_{v_j}(\tilde{\mathbf{x}}_t^u)$. *We have:* $\mathbb{P}\Big[p_1 = \tilde{v}_t^u, \sum_{h=1}^H \big[w_h f_{\tilde{v}_t^u}^h(\tilde{\mathbf{x}}_t^u)\big] \le \gamma\Big] \le C(\mathcal{O}(\epsilon))^\lambda$.
*Proof.*

$$\mathbb{P}\Big[p_1 = \tilde{v}_t^u, \sum_{h=1}^H \big[w_h f_{\tilde{v}_t^u}^h(\tilde{\mathbf{x}}_t^u)\big] \le \gamma\Big]$$

$$\le \mathbb{P}\Big[p_1 = \tilde{v}_t^u, \sum_{h=1}^H w_h \big[\tilde{\eta}_{\tilde{v}_t^u}(\tilde{\mathbf{x}}_t^u) - \epsilon\big] \le \gamma\Big]$$

$$= \mathbb{P}\Big[p_1 = \tilde{v}_t^u, \eta_{\tilde{v}_t^u}(\tilde{\mathbf{x}}_t^u) \ge \eta_{p_2}(\tilde{\mathbf{x}}_t^u), \sum_{h=1}^H w_h \Big[\sum_{v_j \in \mathcal{V}} \tau_{v_j \tilde{v}_t^u}\eta_{v_j}(\tilde{\mathbf{x}}_t^u) - \epsilon\Big] \le \gamma\Big] \qquad (6)$$

$$\le \mathbb{P}\Big[p_1 = \tilde{v}_t^u, \eta_{\tilde{v}_t^u}(\tilde{\mathbf{x}}_t^u) \ge \eta_{p_2}(\tilde{\mathbf{x}}_t^u), \tau_{\tilde{v}_t^u \tilde{v}_t^u}(\tilde{\mathbf{x}}_t^u)\eta_{\tilde{v}_t^u}(\tilde{\mathbf{x}}_t^u) + \sum_{v_j \ne \tilde{v}_t^u} \tau_{v_j \tilde{v}_t^u}(\tilde{\mathbf{x}}_t^u)\eta_{v_j}(\tilde{\mathbf{x}}_t^u) \le \gamma + \epsilon\Big]$$

$$= \mathbb{P}\Big[p_1 = \tilde{v}_t^u, \eta_{p_2}(\tilde{\mathbf{x}}_t^u) \le \eta_{\tilde{v}_t^u}(\tilde{\mathbf{x}}_t^u) \le \frac{\gamma + \epsilon - \sum_{v_j \ne \tilde{v}_t^u} \tau_{v_j \tilde{v}_t^u}(\tilde{\mathbf{x}}_t^u)\eta_{v_j}(\tilde{\mathbf{x}}_t^u)}{\tau_{\tilde{v}_t^u \tilde{v}_t^u}(\tilde{\mathbf{x}}_t^u)}\Big].$$

By replacing $\gamma$ with $\tau_{\tilde{v}_t^u \tilde{v}_t^u}(\tilde{\mathbf{x}}_t^u)\eta_{p_2}(\tilde{\mathbf{x}}_t^u) + \sum_{v_j \ne \tilde{v}_t^u} \tau_{v_j \tilde{v}_t^u}(\tilde{\mathbf{x}}_t^u)\eta_{v_j}(\tilde{\mathbf{x}}_t^u)$, we obtain:

$$\mathbb{P}\Big[p_1 = \tilde{v}_t^u, \sum_{h=1}^H \big[w_h f_{\tilde{v}_t^u}^h(\tilde{\mathbf{x}}_t^u)\big] \le \gamma\Big] \le \mathbb{P}\Big[\eta_{p_2}(\tilde{\mathbf{x}}_t^u) \le \eta_{p_1}(\tilde{\mathbf{x}}_t^u) \le \eta_{p_2}(\tilde{\mathbf{x}}_t^u) + \frac{\epsilon}{\tau_{\tilde{v}_t^u \tilde{v}_t^u}(\tilde{\mathbf{x}}_t^u)}\Big]. \qquad (7)$$

Recall that $\epsilon \le \alpha_0 \tau_{\tilde{v}_t^u \tilde{v}_t^u}(\tilde{\mathbf{x}}_t^u)$, which indicates $\frac{\epsilon}{\tau_{\tilde{v}_t^u \tilde{v}_t^u}(\tilde{\mathbf{x}}_t^u)} \le \alpha_0$. Hence, the relaxed Multiclass Tsybakov Condition holds and the probability is bounded by $C\big(\frac{\epsilon}{\tau_{\tilde{v}_t^u \tilde{v}_t^u}(\tilde{\mathbf{x}}_t^u)}\big)^\lambda$, namely, $C\big(\mathcal{O}(\epsilon)\big)^\lambda$. $\qquad\square$

Note that a small $\epsilon$ relies on large observed data and a powerful $f$, so that (i) the observed data can accurately approximate $\tilde{\eta}$, and (ii) $f$ can closely fit the observed data. Both requirements can be satisfied thanks to the large-scale datasets and deep learning advancements in recommendation.

Theorem 1 indicates that the probability of a reliable target ($p_1 = \tilde{v}_t^u$) keeping low prediction scores across different epochs ($\sum_{h=1}^H \big[w_h f_{\tilde{v}_t^u}^h(\tilde{\mathbf{x}}_t^u)\big] \le \gamma$) is bounded to be low, i.e., no more than $C\big(\mathcal{O}(\epsilon)\big)^\lambda$. This inspires us to rectify unreliable targets by replacing consistently low-scored targets with steadily high-scored items as the following strategy.

**DRUT: Detecting and Replacing Unreliable Targets**. Given an SRS $f$ that is $\epsilon$-close to $\tilde{\eta}$, an instance $\langle \tilde{\mathbf{x}}_t^u, \tilde{v}_t^u \rangle$, the consistently high-scored item $v_m = \arg\max_{v_i \ne \tilde{v}_t^u} \sum_{h=1}^H \big[w_h f_{v_i}^h(\tilde{\mathbf{x}}_t^u)\big]$, and $\beta \in (0, 1]$, we stipulate that if $\sum_{h=1}^H \big[w_h f_{\tilde{v}_t^u}^h(\tilde{\mathbf{x}}_t^u)\big] / \sum_{h=1}^H \big[w_h f_{v_m}^h(\tilde{\mathbf{x}}_t^u)\big] < \beta$, i.e., the target $\tilde{v}_t^u$ is consistently lower-scored than $v_m$ to some extent, then $\tilde{v}_t^u$ should be replaced by $v_m$ in the $H$-th epoch. We denote the output instance of DRUT as $\langle \tilde{\mathbf{x}}_t^u, \hat{v}_t^u \rangle$.

Different from existing methods, DRUT is theoretically error-bounded. Specifically, the error of DRUT, denoted as $\boldsymbol{E}_{\text{DRUT}}$, comes from three cases: (**Case-1**) the true target $p_1$ is $\tilde{v}_t^u$ but is replaced

---

[1] The detailed setting of the weights is related to the self-ensemble mechanism introduced in Section 4.2

by $v_m$. (**Case-2**) the true target $p_1$ is $v_m$ but $\tilde{v}_t^u$ is kept. (**Case-3**) the true target is neither $\tilde{v}_t^u$ nor $v_m$. Correspondingly, $\boldsymbol{E}_{\mathrm{DRUT}}$ can be formulated as below:

$$\boldsymbol{E}_{\mathrm{DRUT}}=\underbrace{\mathbb{P}\left[p_1=\tilde{v}_t^u, p_1\neq v_m, \frac{\sum_{h=1}^H\left[w_h f_{\tilde{v}_t^u}^h(\tilde{\mathbf{x}}_t^u)\right]}{\sum_{h=1}^H\left[w_h f_{v_m}^h(\tilde{\mathbf{x}}_t^u)\right]}<\beta\right]}_{\text{Case-1}}+\underbrace{\mathbb{P}\left[p_1\neq\tilde{v}_t^u, p_1=v_m, \frac{\sum_{h=1}^H\left[w_h f_{\tilde{v}_t^u}^h(\tilde{\mathbf{x}}_t^u)\right]}{\sum_{h=1}^H\left[w_h f_{v_m}^h(\tilde{\mathbf{x}}_t^u)\right]}\geq\beta\right]}_{\text{Case-2}}+\underbrace{\mathbb{P}\left[p_1\neq\tilde{v}_t^u, p_1\neq v_m\right]}_{\text{Case-3}}.$$

Subsequently, Lemma 1 first provides the thresholds $\beta_1$ and $\beta_2$ that can respectively guarantee bounded probabilities for Case-1 and Case-2 of $\boldsymbol{E}_{\mathrm{DRUT}}$. Then in Theorem 2, we prove that even if the chosen $\beta$ deviates from $\beta_1$ and $\beta_2$, DRUT is still error-bounded[2].

**Lemma 1.** *Given Assumption 1 and the set of weights $\left\{w_h \mid 1 \leq h \leq H, 0 \leq w_h \leq 1, \sum_{h=1}^H w_h = 1\right\}$, $\forall\langle\tilde{\mathbf{x}}_t^u, \tilde{v}_t^u\rangle$, assume $\epsilon \leq \min\left[\alpha_0\tau_{\tilde{v}_t^u\tilde{v}_t^u}(\tilde{\mathbf{x}}_t^u), \alpha_0\tau_{v_m v_m}(\tilde{\mathbf{x}}_t^u)\right]$. Let $\beta_1 = \left[\frac{\tau_{\tilde{v}_t^u\tilde{v}_t^u}(\tilde{\mathbf{x}}_t^u)\eta_{p_2}(\tilde{\mathbf{x}}_t^u)+\sum_{v_j\neq\tilde{v}_t^u}\tau_{v_j\tilde{v}_t^u}\eta_{v_j}(\tilde{\mathbf{x}}_t^u)}{\sum_{h=1}^H[w_h f_{v_m}^h(\tilde{\mathbf{x}}_t^u)]}\right]$ and $\beta_2 = \left[\frac{\sum_{h=1}^H[w_h f_{\tilde{v}_t^u}^h(\tilde{\mathbf{x}}_t^u)]}{\tau_{v_m v_m}(\tilde{\mathbf{x}}_t^u)\eta_{p_2}(\tilde{\mathbf{x}}_t^u)+\sum_{v_j\neq v_m}\tau_{v_j v_m}(\tilde{\mathbf{x}}_t^u)\eta_{v_j}(\tilde{\mathbf{x}}_t^u)}\right]$. We have: $\beta \leq \beta_1$ guarantees the probability of Case-1 in $\boldsymbol{E}_{\mathrm{DRUT}}$ is bounded by $C(\mathcal{O}(\epsilon))^\lambda$, and $\beta \geq \beta_2$ guarantees the probability of Case-2 in $\boldsymbol{E}_{\mathrm{DRUT}}$ is bounded by $C(\mathcal{O}(\epsilon))^\lambda$.*

**Theorem 2** (The Upper Bound of $\boldsymbol{E}_{\mathrm{DRUT}}$). *Given Assumption 1 and the set of weights $\left\{w_h \mid 1\leq h \leq H, 0 \leq w_h \leq 1, \sum_{h=1}^H w_h = 1\right\}$, $\forall\langle\tilde{\mathbf{x}}_t^u, \tilde{v}_t^u\rangle$, let $\beta_1 = \left[\frac{\tau_{\tilde{v}_t^u\tilde{v}_t^u}(\tilde{\mathbf{x}}_t^u)\eta_{p_2}(\tilde{\mathbf{x}}_t^u)+\sum_{v_j\neq\tilde{v}_t^u}\tau_{v_j\tilde{v}_t^u}\eta_{v_j}(\tilde{\mathbf{x}}_t^u)}{\sum_{h=1}^H[w_h f_{v_m}^h(\tilde{\mathbf{x}}_t^u)]}\right]$, $\beta_2 = \left[\frac{\sum_{h=1}^H[w_h f_{\tilde{v}_t^u}^h(\tilde{\mathbf{x}}_t^u)]}{\tau_{v_m v_m}(\tilde{\mathbf{x}}_t^u)\eta_{p_2}(\tilde{\mathbf{x}}_t^u)+\sum_{v_j\neq v_m}\tau_{v_j v_m}(\tilde{\mathbf{x}}_t^u)\eta_{v_j}(\tilde{\mathbf{x}}_t^u)}\right]$, $\xi_1 = |\beta - \beta_1|$, and $\xi_2 = |\beta - \beta_2|$. Assume $\xi_2 < \beta_2$, $\epsilon \leq \min\left[\alpha_0\tau_{\tilde{v}_t^u\tilde{v}_t^u}(\tilde{\mathbf{x}}_t^u) - \xi_1, \frac{\alpha_0\tau_{v_m v_m}(\tilde{\mathbf{x}}_t^u)\beta_2(\beta_2-\xi_2)-\xi_2}{\beta_2(\beta_2-\xi_2)}, \frac{1}{2}\left[[\tau_{p_1 p_1}(\tilde{\mathbf{x}}_t^u)-\tau_{p_1 v_m}(\tilde{\mathbf{x}}_t^u)][\alpha_0+\eta_{p_2}(\tilde{\mathbf{x}}_t^u)]-\sum_{v_j\neq p_1}[\tau_{v_j v_m}(\tilde{\mathbf{x}}_t^u)-\tau_{v_j p_1}(\tilde{\mathbf{x}}_t^u)]\eta_{v_j}(\tilde{\mathbf{x}}_t^u)\right]\right]$. We have: $\boldsymbol{E}_{\mathrm{DRUT}} \leq C(\mathcal{O}(\epsilon+\xi_1))^\lambda + C(\mathcal{O}(\epsilon+\xi_2))^\lambda + C(\mathcal{O}(\epsilon))^\lambda$.*

### 3.3 Theorems for Rectifying Unreliable Input

Unreliable items within the input act as mosaics to bewilder SRSs when learning users' true preferences, thus impeding SRSs from predicting the true target. Recall that Theorem 1 suggests consistently low-scored targets are unlikely to be reliable. This inspires us to exploit the prediction scores given by a backward SRS for rectifying unreliable input, that is, deleting consistently low-scored (predicted by a backward SRS) items within the input. To avoid deleting all input items with an unreliable target, we rectify the input of instances that are already processed by DRUT. Formally, given $\langle\tilde{\mathbf{x}}_t^u, \hat{v}_t^u\rangle$, we define a backward SRS $\overleftarrow{f}$ that aims to predict every input item in $\tilde{\mathbf{x}}_t^u$ based on $\overleftarrow{\mathbf{x}}_t^u = \{u, \hat{v}_t^u\}$. Hence, the backward instance for predicting $\tilde{v}_{t-l}^u, (l \in [1, L])$ can be formulated as $\langle\overleftarrow{\mathbf{x}}_t^u, \tilde{v}_{t-l}^u\rangle$. With Assumption 1 being held, and $\eta', \tilde{\eta}', p_1', p_2', \tau', \epsilon'$ respectively being the counterparts of $\eta, \tilde{\eta}, p_1, p_2, \tau, \epsilon$, for backward sequences, we propose the following strategy.

**DDUI: Detecting and Deleting Unreliable Input**. Given $\overleftarrow{f}$ that is $\epsilon'$-close to $\tilde{\eta}'$, $\langle\overleftarrow{\mathbf{x}}_t^u, \tilde{v}_{t-l}^u\rangle$, the consistently high-scored item $\overleftarrow{v}_m = \arg\max_{v_i\neq\tilde{v}_{t-l}^u}\sum_{h=1}^H\left[w_h\overleftarrow{f}_{v_i}^h(\overleftarrow{\mathbf{x}}_t^u)\right]$, and $\beta' \in (0, 1]$, we stipulate that if the ratio $\sum_{h=1}^H\left[w_h\overleftarrow{f}_{\tilde{v}_{t-l}^u}^h(\overleftarrow{\mathbf{x}}_t^u)\right]/\sum_{h=1}^H\left[w_h\overleftarrow{f}_{\overleftarrow{v}_m}^h(\overleftarrow{\mathbf{x}}_t^u)\right] < \beta'$, i.e., $\tilde{v}_{t-l}^u$ is consistently lower-scored than $\overleftarrow{v}_m$ to some extent, then item $\tilde{v}_{t-l}^u$ should be deleted from $\tilde{\mathbf{x}}_t^u$ in the $H$-th epoch. After rectifying the $L$ items in $\tilde{\mathbf{x}}_t^u$, we denote the output instance of DDUI as $\langle\hat{\mathbf{x}}_t^u, \hat{v}_t^u\rangle$.

DDUI differs from DRUT slightly, i.e., we remove unreliable input items to bridge the succeeding and preceding items, instead of replacing them. Since unreliable input items caused by distractions are essentially interruptions in the input sequence, replacing them may introduce new interruptions.

Superior to prior works, DDUI is theoretically guaranteed to work well with bounded error, which comes from two cases: (1) $\tilde{v}_{t-l}^u$ is a reliable input item ($p_1' = \tilde{v}_{t-l}^u$) but is deleted; and (2) $\tilde{v}_{t-l}^u$ is an unreliable input item ($p_1' \neq \tilde{v}_{t-l}^u$) but is kept. Hence, the error of DDUI, denoted as $\boldsymbol{E}_{\mathrm{DDUI}}$, can be correspondingly formulated as the following Eq. 8, and its bound is analyzed in Theorem 3:

$$\boldsymbol{E}_{\mathrm{DDUI}}=\mathbb{P}\left[p_1'=\tilde{v}_{t-l}^u, \frac{\sum_{h=1}^H\left[w_h\overleftarrow{f}_{\tilde{v}_{t-l}^u}^h(\overleftarrow{\mathbf{x}}_t^u)\right]}{\sum_{h=1}^H\left[w_h\overleftarrow{f}_{\overleftarrow{v}_m}^h(\overleftarrow{\mathbf{x}}_t^u)\right]}<\beta'\right]+\mathbb{P}\left[p_1'\neq\tilde{v}_{t-l}^u, \frac{\sum_{h=1}^H\left[w_h\overleftarrow{f}_{\tilde{v}_{t-l}^u}^h(\overleftarrow{\mathbf{x}}_t^u)\right]}{\sum_{h=1}^H\left[w_h\overleftarrow{f}_{\overleftarrow{v}_m}^h(\overleftarrow{\mathbf{x}}_t^u)\right]}\geq\beta'\right]. \quad (8)$$

---

[2]The proofs of Lemma 1 and following theorems can be found in the Appendix.

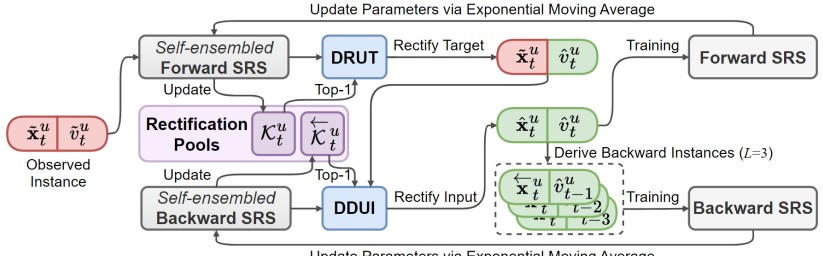

Figure 2: The overall architecture of the proposed BirDRec framework.

**Theorem 3** (The Upper Bound of $E_{\text{DDUI}}$). *Given Assumption 1 and the set of weights* $\{w_h \mid 1 \leq h \leq H, 0 \leq w_h \leq 1, \sum_{h=1}^{H} w_h = 1\}$, $\forall \langle \tilde{\mathbf{x}}_t^u, \tilde{v}_{t-l}^u \rangle$, *let* $\beta_1' = \left[ \frac{\tau'_{\tilde{v}_{t-l}^u \tilde{v}_{t-l}^u}(\tilde{\mathbf{x}}_t^u) \eta'_{p_2'}(\tilde{\mathbf{x}}_t^u) + \sum_{v_j \neq \tilde{v}_{t-l}^u} \tau'_{v_j \tilde{v}_{t-l}^u} \eta'_{v_j}(\tilde{\mathbf{x}}_t^u)}{\sum_{h=1}^{H}[w_h \bar{f}_{\bar{v}_m}^h(\tilde{\mathbf{x}}_t^u)]} \right]$,
$\beta_2' = \left[ \frac{\sum_{h=1}^{H}[w_h \bar{f}_{\tilde{v}_{t-l}^u}^h(\tilde{\mathbf{x}}_t^u)]}{\tau'_{\bar{v}_m \bar{v}_m}(\tilde{\mathbf{x}}_t^u) \eta'_{p_2'}(\tilde{\mathbf{x}}_t^u) + \sum_{v_j \neq \bar{v}_m} \tau'_{v_j \bar{v}_m}(\tilde{\mathbf{x}}_t^u) \eta'_{v_j}(\tilde{\mathbf{x}}_t^u)} \right]$, $\xi_1' = |\beta' - \beta_1'|$, $\xi_2' = |\beta' - \beta_2'|$. *Assume* $\xi_2' < \beta_2'$,
$\epsilon' \leq \min\left[ \alpha_0 \tau'_{\tilde{v}_{t-l}^u \tilde{v}_{t-l}^u}(\tilde{\mathbf{x}}_t^u) - \xi_1', \frac{\alpha_0 \tau'_{\bar{v}_m \bar{v}_m}(\tilde{\mathbf{x}}_t^u) \beta_2'(\beta_2' - \xi_2') - \xi_2'}{\beta_2'(\beta_2' - \xi_2')}, \frac{1}{2}\left[ [\tau'_{p_1' p_1'}(\tilde{\mathbf{x}}_t^u) - \tau'_{p_1' \bar{v}_m}(\tilde{\mathbf{x}}_t^u)][\alpha_0 + \eta'_{p_2'}(\tilde{\mathbf{x}}_t^u)] - \sum_{v_j \neq p_1'} [\tau'_{v_j \bar{v}_m}(\tilde{\mathbf{x}}_t^u) - \tau'_{v_j p_1'}(\tilde{\mathbf{x}}_t^u)] \eta'_{v_j}(\tilde{\mathbf{x}}_t^u)] \right]$. *We have:* $E_{\text{DDUI}} \leq C(\mathcal{O}(\epsilon' + \xi_1'))^\lambda + C(\mathcal{O}(\epsilon' + \xi_2'))^\lambda + C(\mathcal{O}(\epsilon'))^\lambda$.

## 4   The Proposed BirDRec Framework

By integrating DRUT and DDUI into existing SRSs, we introduce `BirDRec`, a model-agnostic bidirectional data rectification framework, which can rectify both unreliable targets and input with theoretical guarantees for more robust SRSs. Yet, the complexity of BirDRec is prohibitively high due to the calculation and storage of prediction scores for each instance across different epochs in DRUT and DDUI. To ease this issue, we devise a rectification sampling strategy that avoids prediction on the full item set to replace unreliable targets or delete unreliable input, thereby reducing the time complexity of BirDRec. Meanwhile, we adopt the self-ensemble mechanism [14] to approximate the weighted average prediction scores of different epochs, thus avoiding preserving scores of all epochs and reducing the space complexity.

**Framework Overview**. Accordingly, the efficiency-improved BirDRec is depicted in Fig. 2. Specifically, BirDRec first leverages the self-ensembled forward SRS to rectify the target of an instance via DRUT, and then the input is rectified by the self-ensembled backward SRS via DDUI. Thereafter, the rectified instance and its $L$ backward instances are respectively used to train the forward and backward SRSs, which are finally employed to update the corresponding self-ensembled SRSs.

### 4.1   Reducing Time Complexity via Rectification Sampling

The time complexity of BirDRec is primarily attributed to the search for consistently high-scored items ($v_m$ and $\bar{v}_m$), which requires the calculation of prediction scores between every instance and all items. Concretely, the per-epoch time complexity of BirDRec is $\mathcal{O}(N \cdot (|\mathcal{V}| + L))$, where $N$ denotes the number of instances. To mitigate this substantial computational burden, we propose a rectification sampling strategy that circumvents the need to search the entire item set.

**Rectification Sampling**. Each instance $\langle \tilde{\mathbf{x}}_t^u, \tilde{v}_t^u \rangle$ is assigned two rectification pools $\mathcal{K}_t^u$ and $\overleftarrow{\mathcal{K}}_t^u$ to rectify the target and input, respectively. At first, $\mathcal{K}_t^u$ is initialized by the $K$ succeeding items of $\tilde{v}_t^u$, i.e., $[\tilde{v}_{t+1}^u, ..., \tilde{v}_{t+K}^u]$, which are potentially better substitutions for $\tilde{v}_t^u$. Similarly, $\overleftarrow{\mathcal{K}}_t^u$ is initialized with the $K$ preceding items of $\tilde{v}_{t-1}^u$. Next, in each epoch, the items in $\mathcal{K}_t^u$, together with $K$ additional items that are randomly sampled from $\mathcal{V}$, are ranked in descending order w.r.t. their weighted average prediction scores over different epochs (i.e., $\sum_{h=1}^{H}[w_h f_{v_i}^h(\tilde{\mathbf{x}}_t^u)]$). Then the top-1 item, denoted as $\hat{v}_m$, is adopted as the approximation of $v_m$ in DRUT, while the top-$K$ items are retained to update $\mathcal{K}_t^u$ for the next epoch. The rationale behind approximating $v_m$ with $\hat{v}_m$ is supported by Theorem 4, which indicates that the relative rank of an item over a list of randomly sampled items can approximate this item's relative rank over the full item set. That is, the top-1 item $\hat{v}_m$ tends to be ranked highly over

Table 1: Statistics of the datasets.

| Dataset | # Users | # Items | # Interactions | Avg. Length | Sparsity |
|---|---|---|---|---|---|
| ML-1M (ML) | 6,040 | 3,417 | 999,611 | 165.5 | 95.16% |
| Beauty (Be) | 22,362 | 12,102 | 198,502 | 8.9 | 99.93% |
| Yelp (Ye) | 22,844 | 16,552 | 236,999 | 10.4 | 99.94% |
| QK-Vedio (QK) | 30,704 | 41,534 | 2,268,935 | 73.9 | 99.82% |

$\mathcal{V}$. The updating rules of $\overleftarrow{\mathcal{K}}_t^u$ for DDUI is defined similarly as $\mathcal{K}_t^u$. As a result, the per-epoch time complexity of BirDRec is reduced from $\mathcal{O}\big(N \cdot (|\mathcal{V}| + L)\big)$ to $\mathcal{O}\big(N \cdot (K + L)\big)$.

**Theorem 4.** *Let $\mathcal{R}$ be a list of $K$ items randomly sampled from $\mathcal{V}$ with replacement, $\zeta \in (0,1)$, $r_H(\mathbf{x}_t^u, v_i) = \sum_{v_j \in \mathcal{V}} \mathbb{I}\big[\sum_{h=1}^H [w_h f_{v_j}^h(\tilde{\mathbf{x}}_t^u)] > \sum_{h=1}^H [w_h f_{v_i}^h(\tilde{\mathbf{x}}_t^u)]\big]$ be the rank of item $v_i$ over the entire item set at the $H$-th epoch, and $\hat{r}_H(\mathbf{x}_t^u, v_i) = \sum_{v_j \in \mathcal{R}} \mathbb{I}\big[\sum_{h=1}^H [w_h f_{v_j}^h(\tilde{\mathbf{x}}_t^u)] > \sum_{h=1}^H [w_h f_{v_i}^h(\tilde{\mathbf{x}}_t^u)]\big]$ be the rank of $v_i$ over $\mathcal{R}$ at the $H$-th epoch. We have: $\mathbb{P}\big[\big|\frac{\hat{r}_H(\mathbf{x}_t^u, v_i)}{K} - \frac{r_H(\mathbf{x}_t^u, v_i)}{|\mathcal{V}|}\big| \geq \zeta\big] \leq \exp(-2K\zeta^2)$.*

### 4.2 Reducing Space Complexity via Self-ensemble Mechanism

The huge space cost of BirDRec is caused by the storage of prediction scores between each instance and all items in every epoch, for the sake of calculating the weighted average prediction scores.

Specifically, at the $H$-th epoch, the space complexity of preserving all prediction scores is $\mathcal{O}(N \cdot |\mathcal{V}| \cdot H)$. To save space, we thus approximate the weighted average scores with the self-ensemble mechanism [14], thereby avoiding storing prediction scores of different epochs. Formally, let $f^H$ be an SRS parameterized by $\boldsymbol{\theta}_H$, and $\overline{f}^H$ be a self-ensembled SRS parameterized by $\overline{\boldsymbol{\theta}}_H = \sum_{h=1}^H w_h \boldsymbol{\theta}_h$. It has proven that [14] the difference between the weighted average prediction scores $(\sum_{h=1}^H w_h f_{v_i}^h(\mathbf{x}_t^u))$ and the prediction of the self-ensembled SRS $(\overline{f}_{v_i}^H(\mathbf{x}_t^u))$ is of the second order of smallness, if and only if $w_h = \rho^{H-h}(1-\rho)^{1-\delta(h-1)}$, where $\rho \in (0,1)$ denotes the exponential decay rate for ensembling; and $\delta(\cdot)$ is the unit impulse function, i.e., $\delta(0) = 1$, otherwise 0. With such self-ensemble, there is no need to store SRSs of each epoch, as $\overline{\boldsymbol{\theta}}_H$ can be efficiently derived from $\overline{\boldsymbol{\theta}}_{H-1}$ and $\boldsymbol{\theta}_H$ with the exponential moving average as follows:

$$\overline{\boldsymbol{\theta}}_H = \sum_{h=1}^H \rho^{H-h}(1-\rho)^{1-\delta(h-1)}\boldsymbol{\theta}_h = \rho\overline{\boldsymbol{\theta}}_{H-1} + (1-\rho)\boldsymbol{\theta}_H. \tag{9}$$

By doing so, the burden of retaining prediction scores of different epochs is reduced to maintaining an extra self-ensembled SRS. As the parameters of an SRS mainly consist of the user and item embeddings, the per-epoch space complexity of BirDRec is reduced from $\mathcal{O}\big((|\mathcal{U}| + |\mathcal{V}|) \cdot d + (L + |\mathcal{V}| \cdot H) \cdot N\big)$ to $\mathcal{O}\big((|\mathcal{U}| + |\mathcal{V}|) \cdot d + (L + K) \cdot N\big)$, where $d$ represents the embedding size. Furthermore, to reduce the number of parameters and mitigate overfitting, the (self-ensembled) forward and backward SRSs in BirDRec share the same user and item embeddings.

## 5 Experiments and Results

### 5.1 Experimental Settings

**Datasets.** We adopt four real-world datasets with varying domains, sizes, sparsity, and average sequence lengths shown in Table 1. Specifically, `ML-1M (ML)`[45] is a popular movie recommendation benchmark. `Beauty (Be)` [46] is the product review dataset collected from Amazon.com. `Yelp (Ye)` [10] is a business recommendation dataset released by Yelp.com. `QK-Video (QK)` [47] is a video recommendation dataset crawled from Tencent.com. Following [3, 8, 9], we preprocess all datasets by removing users and items whose interactions are less than 5.

**Baselines.** To verify the generality of BirDRec, we implement it with *vanilla SRSs* based on representative backbones. In particular, `FPMC` [15] is based on Markov Chain. `GRU4Rec`[17], `Caser` [20], and `MAGNN` [27] are built on RNN, CNN, and GNN, respectively. `SASRec` [28] and `BERT4Rec` [29] are based on Transformer. Meanwhile, to validate its effectiveness and efficiency, we compare BirDRec with state-of-the-art *robust SRSs* including `BERD` [8], `FMLP-Rec` [9], and `STEAM` [10], which aim to tackle unreliable targets, unreliable input, and both, respectively.

**Evaluation Protocol.** Following [9, 48, 49], three widely-used metrics are adopted to evaluate the ranking quality, namely, HR, NDCG, and MRR. For all these metrics, higher metric values suggest

Table 2: Performance comparison with vanilla SRSs trained with the original 'Plain' setting and our BirDRec framework. *Improv.* means the relative improvement of BirDRec over the 'Plain' setting. The significance of the improvement is determined by a paired t-test with $p \leq 0.001$.

| Datasets | | ML-1M | | | | | Beauty | | | | |
|---|---|---|---|---|---|---|---|---|---|---|---|
| Backbones | Settings | HR@5 | HR@10 | NDCG@5 | NDCG@10 | MRR | HR@5 | HR@10 | NDCG@5 | NDCG@10 | MRR |
| FPMC | Plain | 0.1317 | 0.2087 | 0.0821 | 0.1069 | 0.0929 | 0.0315 | 0.0527 | 0.0201 | 0.0259 | 0.0244 |
| | BirDRec | **0.1534** | **0.2407** | **0.0972** | **0.1251** | **0.1059** | **0.0513** | **0.0750** | **0.0348** | **0.0423** | **0.0379** |
| | *Improv.* | 16.48% | 15.33% | 18.39% | 17.03% | 13.99% | 62.86% | 42.31% | 73.13% | 63.32% | 55.33% |
| Caser | Plain | 0.1592 | 0.2450 | 0.1023 | 0.1272 | 0.1099 | 0.0377 | 0.0574 | 0.0245 | 0.0308 | 0.0279 |
| | BirDRec | **0.2091** | **0.2872** | **0.1426** | **0.1666** | **0.1443** | **0.0445** | **0.0638** | **0.0307** | **0.0369** | **0.0334** |
| | *Improv.* | 31.34% | 17.22% | 39.39% | 30.97% | 31.30% | **18.04%** | **11.15%** | **25.31%** | **19.81%** | 19.71% |
| GRU4Rec | Plain | 0.1597 | 0.2427 | 0.1044 | 0.1304 | 0.1126 | 0.0293 | 0.0470 | 0.0186 | 0.0243 | 0.0228 |
| | BirDRec | **0.2167** | **0.2973** | **0.1495** | **0.1751** | **0.1511** | **0.0526** | **0.0761** | **0.0357** | **0.0434** | **0.0386** |
| | *Improv.* | 35.69% | 22.50% | 43.20% | 34.28% | 34.19% | 79.52% | 61.91% | 91.94% | 78.60% | 69.30% |
| SASRec | Plain | 0.1769 | 0.2662 | 0.1168 | 0.1451 | 0.1252 | 0.0475 | 0.0691 | 0.0313 | 0.0385 | 0.0343 |
| | BirDRec | **0.2352** | **0.3259** | **0.1631** | **0.1915** | **0.1647** | **0.0653** | **0.0903** | **0.0459** | **0.0537** | **0.0482** |
| | *Improv.* | 32.96% | 22.43% | 39.64% | 31.98% | 31.55% | 37.47% | 30.68% | 46.65% | 39.48% | 40.52% |
| BERT4Rec | Plain | 0.1736 | 0.2655 | 0.1142 | 0.1429 | 0.1219 | 0.0356 | 0.0563 | 0.0231 | 0.0293 | 0.0271 |
| | BirDRec | **0.2319** | **0.3152** | **0.1613** | **0.1871** | **0.1622** | **0.0649** | **0.0896** | **0.0466** | **0.0544** | **0.0489** |
| | *Improv.* | 33.58% | 18.72% | 41.24% | 30.93% | 33.06% | 82.30% | 59.15% | 101.73% | 85.67% | 80.44% |
| MAGNN | Plain | 0.1802 | 0.2692 | 0.1196 | 0.1479 | 0.1265 | 0.0566 | 0.0798 | 0.0380 | 0.0455 | 0.0403 |
| | BirDRec | **0.2341** | **0.3211** | **0.1628** | **0.1903** | **0.1649** | **0.0634** | **0.0890** | **0.0435** | **0.0518** | **0.0456** |
| | *Improv.* | 29.91% | 19.28% | 36.12% | 28.67% | 30.36% | 12.01% | 11.53% | 14.47% | 13.85% | 13.15% |

| Datasets | | Yelp | | | | | QK-Vedio | | | | |
|---|---|---|---|---|---|---|---|---|---|---|---|
| FPMC | Plain | 0.0502 | 0.0754 | 0.0398 | 0.0431 | 0.0425 | 0.0483 | 0.0803 | 0.0303 | 0.0405 | 0.0384 |
| | BirDRec | **0.0743** | **0.0914** | **0.063** | **0.0686** | **0.0673** | **0.0756** | **0.1193** | **0.0483** | **0.0624** | **0.0569** |
| | *Improv.* | 48.01% | 21.22% | 58.29% | 59.16% | 58.35% | 56.52% | 48.57% | 59.41% | 54.07% | 48.18% |
| Caser | Plain | 0.0337 | 0.0519 | 0.0228 | 0.0286 | 0.0277 | 0.0498 | 0.0831 | 0.0314 | 0.0396 | |
| | BirDRec | **0.0651** | **0.0818** | **0.0519** | **0.0571** | **0.0548** | **0.0746** | **0.1221** | **0.0479** | **0.0632** | **0.0573** |
| | *Improv.* | 93.18% | 57.61% | 127.63% | 99.65% | 97.83% | 49.80% | 46.93% | 52.55% | 50.12% | 44.70% |
| GRU4Rec | Plain | 0.0320 | 0.0530 | 0.0198 | 0.0265 | 0.0257 | 0.0485 | 0.0816 | 0.0305 | 0.0410 | 0.0388 |
| | BirDRec | **0.0741** | **0.0921** | **0.0602** | **0.0659** | **0.0638** | **0.0776** | **0.1253** | **0.0497** | **0.0651** | **0.0591** |
| | *Improv.* | 131.56% | 73.77% | 204.04% | 148.68% | 148.25% | 60.00% | 53.55% | 62.95% | 58.78% | 52.32% |
| SASRec | Plain | 0.0404 | 0.0574 | 0.0295 | 0.0349 | 0.0341 | 0.0511 | 0.0858 | 0.0326 | 0.0435 | 0.0408 |
| | BirDRec | **0.0771** | **0.0965** | **0.0626** | **0.0687** | **0.0663** | **0.0815** | **0.1306** | **0.0523** | **0.0682** | **0.0616** |
| | *Improv.* | 90.84% | 68.12% | 112.20% | 96.85% | 94.43% | 59.49% | 52.21% | 60.43% | 56.78% | 50.98% |
| BERT4Rec | Plain | 0.0421 | 0.0597 | 0.0318 | 0.0375 | 0.0368 | 0.0558 | 0.0925 | 0.0354 | 0.0473 | 0.0442 |
| | BirDRec | **0.0734** | **0.0918** | **0.0597** | **0.0657** | **0.0635** | **0.0817** | **0.1297** | **0.0524** | **0.0677** | **0.0613** |
| | *Improv.* | 74.35% | 53.77% | 87.74% | 75.20% | 72.55% | 46.42% | 40.22% | 48.02% | 43.13% | 38.69% |
| MAGNN | Plain | 0.0436 | 0.0602 | 0.0337 | 0.0388 | 0.0383 | 0.0529 | 0.0880 | 0.0344 | 0.0452 | 0.0426 |
| | BirDRec | **0.0560** | **0.0792** | **0.0413** | **0.0488** | **0.0465** | **0.0742** | **0.1184** | **0.0479** | **0.0619** | **0.0565** |
| | *Improv.* | 28.44% | 31.56% | 22.55% | 25.77% | 21.41% | 40.26% | 34.55% | 39.24% | 36.95% | 32.63% |

(a) Time per Epoch    (b) Convergence Epoch    (c) Memory Usage    (d) GPU Memory Usage

Figure 3: The comparison regarding time and storage costs.

better recommendation accuracy. For each user, we preserve the last two interactions for validation and testing, while the rest are used for training. The training of each model is carried out five times to report the average results. During testing, we follow [3, 9, 27] and evaluate the ranking results over the whole item set for fair comparison [50].

**Implementation Details.** For all methods, Xavier initializer [51] and Adam optimizer [52] are adopted; and the best hyper-parameter settings are empirically found based on the performance on the validation set. For BirDRec, it is implemented by PyTorch with batch_size = 1024, $d = 64$, learning_rate = 0.01 for Yelp and 0.001 for other datasets, $L = 5$, $\beta = 0.1$, $\beta' = 0.1$, $K = 10$, and $\rho = 0.9$. To ensure accurate rectification with reduced $\epsilon$, BirDRec is trained without rectification in the first 10 epochs. All the experiments are conducted on an NVIDIA Quadro RTX 8000 GPU[3].

## 5.2 Experimental Results and Analysis

**Overall Comparison**. Table 2 presents the performance of vanilla SRSs built on various representative backbones. Each SRS is trained under two different settings: the original Plain setting and our

---

[3]Our source code and experiment details (e.g., parameter settings for baselines) are in the Appendix.

Table 3: Performance comparison with existing robust SRSs, where the best performance is boldfaced and the runner up is marked by '*'. *Improv.* means the relative improvement of BirDRec over the runner up. The significance of the improvement is determined by a paired t-test with $p \leq 0.001$.

| Datasets | ML-1M | | | | | Beauty | | | | |
|---|---|---|---|---|---|---|---|---|---|---|
| Metrics | HR@5 | HR@10 | NDCG@5 | NDCG@10 | MRR | HR@5 | HR@10 | NDCG@5 | NDCG@10 | MRR |
| BERD | 0.1922* | 0.2814* | 0.1267* | 0.1554* | 0.1335* | 0.0507* | 0.0745 | 0.0332 | 0.0406 | 0.0364 |
| FMLP-Rec | 0.1789 | 0.2685 | 0.1207 | 0.1492 | 0.1299 | 0.0501 | 0.0743 | 0.0384* | 0.0448* | 0.0404* |
| STEAM | 0.1198 | 0.1950 | 0.0765 | 0.1006 | 0.0874 | 0.0495 | 0.0765* | 0.0324 | 0.0414 | 0.0371 |
| BirDRec | **0.2352** | **0.3259** | **0.1631** | **0.1915** | **0.1647** | **0.0653** | **0.0903** | **0.0459** | **0.0537** | **0.0482** |
| *Improv.* | 22.37% | 15.81% | 28.73% | 23.23% | 23.37% | 30.34% | 18.04% | 19.53% | 19.87% | 19.31% |

| Datasets | Yelp | | | | | QK-Vedio | | | | |
|---|---|---|---|---|---|---|---|---|---|---|
| BERD | 0.0423 | 0.0636 | 0.0304 | 0.0360 | 0.0351 | 0.0548 | 0.0958 | 0.0352 | 0.0496 | 0.0461 |
| FMLP-Rec | 0.0499 | 0.0702 | 0.0374 | 0.0433 | 0.0422 | 0.0604* | 0.0991 | 0.0391* | 0.0514* | 0.0480* |
| STEAM | 0.0556* | 0.0822* | 0.0387* | 0.0473* | 0.0448* | 0.0597 | 0.1021* | 0.0371 | 0.0507 | 0.0465 |
| BirDRec | **0.0771** | **0.0965** | **0.0626** | **0.0687** | **0.0663** | **0.0815** | **0.1306** | **0.0523** | **0.0682** | **0.0616** |
| *Improv.* | 38.67% | 17.40% | 61.76% | 45.24% | 47.99% | 34.93% | 27.91% | 33.76% | 32.68% | 28.33% |

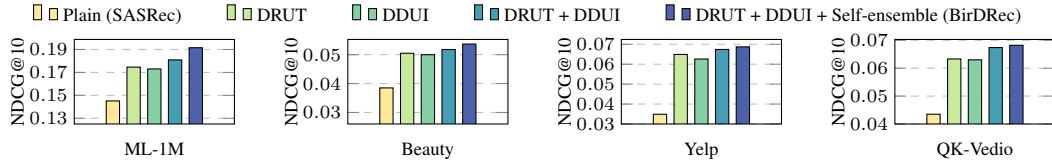

Figure 4: Ablation study on different variants of BirDRec.

BirDRec framework. We can note that all the baselines are boosted significantly on all datasets with the aid of BirDRec, which demonstrates the ***generality*** of BirDRec.

Table 3 compares BirDRec with state-of-the-art robust SRSs. Considering BERD and STEAM are both built upon Transformer, we also adopt a transformer-based SRS, SASRec, as the default backbone of BirDRec in the following experiments to ensure fair comparisons. The results show that BirDRec dramatically outperforms existing robust SRSs, confirming the ***effectiveness*** of our error-bounded rectification strategies in comparison to existing methods that lack theoretical guarantees.

Fig. 3 compares the time and storage cost of BirDRec and STEAM, both of which aim to handle unreliable input and targets simultaneously. In particular, Figs. 3(a) and (b) show that BirDRec executes $4.28$ times faster than STEAM in each epoch and converges with fewer epochs. Meanwhile, Figs. 3(c) and (d) show that BirDRec incurs only half the storage cost of STEAM on most datasets. These results highlight the ***efficiency*** of our sampling strategy and the self-ensemble mechanism. The storage cost of BirDRec is marginally higher than STEAM's on ML-1M. This is because the advantage of our rectification sampling strategy is less obvious on datasets with small item sets (see Table 1), indicating the superior scalability of our BirDRec on large-scale item sets.

**Ablation Study**. To check the efficacy of essential strategies of BirDRec (DRUT, DDUI, and the self-ensemble), we add these strategies incrementally to a plain representative SRS – SASRec. Note that without self-ensemble, DRUT and DDUI only consider the prediction scores at the latest epoch for data rectification. The results are presented in Fig. 4 and similar trends can be noted with the rest SRSs on the other metrics. First, adding either DRUT or DDUI to SASRec brings dramatic improvements, implying the effectiveness of DRUT and DDUI in rectifying unreliable targets and input, respectively. Second, using DRUT and DDUI together is better than leveraging each of them solely, indicating the necessity of rectifying both unreliable targets and input for more robust SRSs. Moreover, adding the self-ensemble mechanism can further boost the accuracy, which confirms the efficacy of considering prediction scores of different epochs for data rectification. Overall, the performance gain of separately adding DRUT, DDUI, and self-ensemble is $45.7\%$, $43.3\%$, and $3.1\%$, respectively. That is, DRUT and DDUI contribute more than the self-ensemble to our BirDRec.

**Hyper-parameter Analysis.** We analyze the impact of key hyper-parameters, i.e., the rectification thresholds $\beta$, $\beta'$, the size $K$ of the rectification pool, and the exponential decay rate $\rho$. The results are presented in Fig. 5 with several major findings (similar trends can be noted with the other metrics on the rest datasets). (i) The best choice for $\beta$ and $\beta'$ is $0.1$, showing that small thresholds tend to maintain unreliable instances while large ones may mistakenly rectify reliable data. (ii) $K \geq 10$ is sufficient to gain better accuracy. (iii) $\rho \geq 0.5$ yields better performance, that is, it is beneficial to consider predictions in earlier epochs. (iv) Even if the hyper-parameters of BirDRec are not optimally set, BirDRec still dramatically outperforms the best baseline (BERD) on ML-1M.

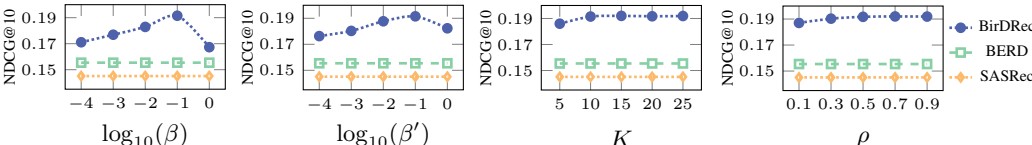

Figure 5: Effect of key hyper-parameters $\beta$, $\beta'$, $K$, and $\rho$ of BirDRec on ML-1M dataset.

**Case Study.** Fig. 6 presents a real instance that is rectified by BirDRec on ML-1M. Specifically, in the DRUT module, the target movie 'Schindler's List' consistently gets lower scores than the movie 'Terminator 2' in the rectification pool, and is then replaced by 'Terminator 2'. This is reasonable, given that 'Terminator 2' aligns more closely with the input Sci-fi movies than 'Schindler's List'. Subsequently, in the DDUI module, the input movie 'American Beauty' is consistently lower-scored than the movie 'Star Wars IV' in the rectification pool and is thus deleted. This decision is justifiable, considering that the rectified target, 'Terminator 2', generally lacks relevance to the input 'American Beauty' across various aspects, including genres, actors, directors, tags, etc.

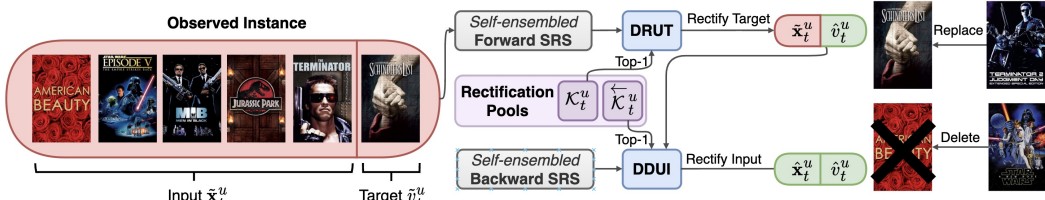

Figure 6: The overall architecture of the proposed BirDRec framework.

**Limitations of BirDRec.** Although BirDRec exhibits its superiority through our extensive experiments, its limitations are two-fold. Firstly, the improvement of BirDRec on storage cost is limited for smaller item sets. This is primarily because the rectification sampling strategy with self-ensemble has less apparent advantages on datasets with smaller item sets. To be specific, the storage cost reduction for calculating weighted average prediction scores is from $O(|V| * H)$ to $O(K)$ for each instance. Thus if $|V|$ is small, the benefit of this reduction will be less obvious. Secondly, although BirDRec is significantly faster than the latest robust SRS (STEAM), it is worth noting that BirDRec is 1.6 times on average slower than its backbone model (as depicted in Fig. 3) in each training epoch. This increased training time could be a practical concern in systems with extremely large-scale datasets and real-time recommendation demands.

## 6   Conclusion

This work, *for the first time*, provides theoretically guaranteed data rectification strategies to tackle both unreliable input and targets for more robust SRSs. The proposed strategies are further integrated into a model-agnostic bidirectional data rectification framework, BirDRec, that can be flexibly implemented with most existing SRSs, for robust training against unreliable data. Additionally, we devise a rectification sampling strategy to reduce the computational cost of BirDRec; meanwhile, a self-ensemble mechanism is adopted to reduce the space complexity. Extensive experiments verify the generality, effectiveness, and efficiency of the proposed BirDRec.

## Acknowledgments

The work is partially supported by the National Key Research and Development Program of China (2020YFB1707900), the National Natural Science Foundation of China (Nos. U22A2025, 62072088, 62232007), Liaoning Provincial Science and Technology Plan Project - Key R&D Department of Science and Technology (No. 2023JH2/101300182), and ARC Discovery Projects DP200101441 and DP230100676.

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

# Theoretically Guaranteed Bidirectional Data Rectification for Robust Sequential Recommendation
# – Appendix –

This Appendix is divided into three sections. First, Section A empirically validates the feasibility of Assumption 1. Next, in Section B, complete proofs of all the lemmas and theorems are presented. Finally, in Section C, we provide detailed settings of baselines and additional experimental results, including the hyper-parameter analysis and the percentage of rectified data. Our code is available at: https://github.com/AlchemistYT/BirDRec.

## A    The Feasibility of Assumption 1

**Assumption 1.** *The users' true preference distribution $\eta$ fulfills the relaxed Multiclass Tsybakov Condition [1] with constants $C > 0$, $\lambda > 0$, and $\alpha_0 \in (0, 1]$, such that for all $\alpha \in (0, \alpha_0]$,*

$$\mathbb{P}\big[\eta_{p_1}(\mathbf{x}_t^u) - \eta_{p_2}(\mathbf{x}_t^u) \le \alpha\big] \le C\alpha^{\lambda}. \tag{1}$$

The feasibility of Assumption 1 relies on large $C$ and small $\lambda$. In order to estimate the values of $C$ and $\lambda$, the initial step is to approximate the true preference distribution $\eta$. To achieve this, we first obtain a reliable dataset $\mathcal{D}$ via the heuristic method proposed by [2]. The heuristic method measures the matching degree between the input and target of each instance from two aspects: item co-occurrence and item properties, then those instances with matching degrees lower than a threshold are filtered as unreliable data. We use a large threshold (0.9) to filter the dataset rigorously, ensuring the vast majority of the maintained instances in $\mathcal{D}$ are reliable. Subsequently, as suggested by [3], we train a classic SRS, SASRec [4], on the filtered reliable dataset $\mathcal{D}$ and then use the prediction of SASRec to approximate $\eta$. Formally, for each reliable instance $\langle \mathbf{x}_t^u, v_t^u \rangle$ from the reliable dataset $\mathcal{D}$, the SRS prediction score $f_{v_t^u}(\mathbf{x}_t^u)$ is employed as the approximation of $\eta_{v_t^u}(\mathbf{x}_t^u)$.

Next, we densely sample $\alpha$ from 0.05 to 0.9 with step size 0.005 and calculate the corresponding left-hand-side probability of Eq. 1 with the following relative frequency $F_\alpha$, and collect a series of $\big(\log(\alpha), \log(F_\alpha)\big)$ data points:

$$F_\alpha = \frac{1}{|\mathcal{D}|} \sum_{\langle \mathbf{x}_t^u, v_t^u \rangle \in \mathcal{D}} \mathbb{I}\big[f_{\hat{p}_1}(\mathbf{x}_t^u) - f_{\hat{p}_2}(\mathbf{x}_t^u) \le \alpha\big], \tag{2}$$

where $\hat{p}_1$ and $\hat{p}_2$ are respectively the top- and middle-ranked items according to $f$, namely, $\sum_{v_i \in \mathcal{V}} \mathbb{I}\big[f_{\hat{p}_1}(\mathbf{x}_t^u) \ge f_{v_i}(\mathbf{x}_t^u)\big] = |\mathcal{V}|$, $\sum_{v_i \in \mathcal{V}} \mathbb{I}\big[f_{\hat{p}_2}(\mathbf{x}_t^u) \ge f_{v_i}(\mathbf{x}_t^u)\big] = \lfloor |\mathcal{V}|/2 \rfloor$. We then use $\log(F_\alpha)$ to approximate $\log(C\alpha^\lambda)$. As shown by the blue dots in Fig. 6, the collected $\big(\log(\alpha), \log(F_\alpha)\big)$ data points are generally linearly distributed, which allows us to estimate $C$ and $\lambda$ with linear regression according to $\log(C\alpha^\lambda) = \log(C) + \lambda \log(\alpha)$. As a result, Fig. 6 shows that the estimated $C$ and $\lambda$ are respectively restricted in $(0.55, 0.70)$ and $(1.37, 4.01)$ on real-world datasets from various domains, which validates the feasibility of Assumption 1.

37th Conference on Neural Information Processing Systems (NeurIPS 2023).

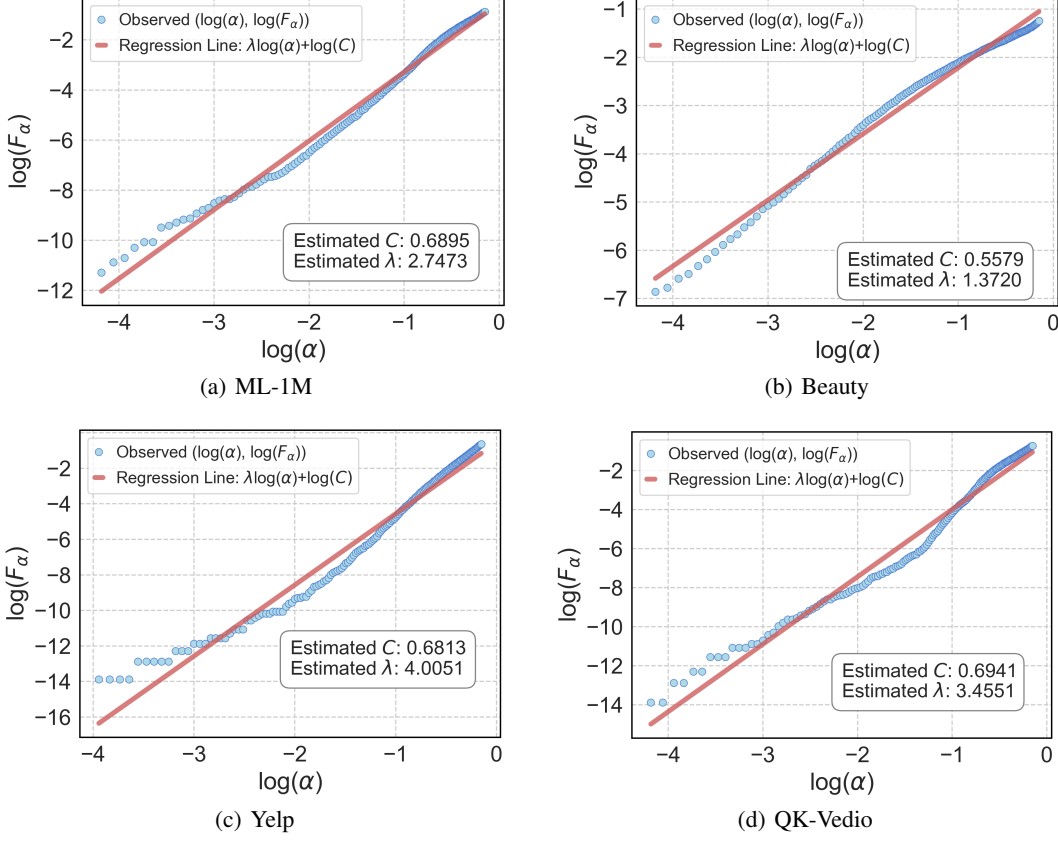

Figure 1: The estimated constants $C$ and $\lambda$ on various datasets.

(a) ML-1M
(b) Beauty
(c) Yelp
(d) QK-Vedio

# B  Proofs of Lemmas and Theorems

## B.1  The Proof of Theorem 1

**Theorem 1.** *Given Assumption 1, let $\{w_h \mid 1 \leq h \leq H, 0 \leq w_h \leq 1, \sum_{h=1}^{H} w_h = 1\}$ be the weights for averaging prediction scores of different epochs. $\forall \langle \tilde{\mathbf{x}}_t^u, \tilde{v}_t^u \rangle$, assume $\epsilon \leq \alpha_0 \tau_{\tilde{v}_t^u \tilde{v}_t^u}(\tilde{\mathbf{x}}_t^u)$. Let $\gamma = \tau_{\tilde{v}_t^u \tilde{v}_t^u}(\tilde{\mathbf{x}}_t^u)\eta_{p_2}(\tilde{\mathbf{x}}_t^u) + \sum_{v_j \neq \tilde{v}_t^u} \tau_{v_j \tilde{v}_t^u}(\tilde{\mathbf{x}}_t^u)\eta_{v_j}(\tilde{\mathbf{x}}_t^u)$. We have: $\mathbb{P}\left[p_1 = \tilde{v}_t^u, \sum_{h=1}^{H} \left[w_h f_{\tilde{v}_t^u}^h(\tilde{\mathbf{x}}_t^u)\right] \leq \gamma\right] \leq C(\mathcal{O}(\epsilon))^\lambda$.*

*Proof.*

$$\mathbb{P}\left[p_1 = \tilde{v}_t^u, \sum_{h=1}^{H} \left[w_h f_{\tilde{v}_t^u}^h(\tilde{\mathbf{x}}_t^u)\right] \leq \gamma\right]$$

$$\leq \mathbb{P}\left[p_1 = \tilde{v}_t^u, \sum_{h=1}^{H} w_h \left[\tilde{\eta}_{\tilde{v}_t^u}(\tilde{\mathbf{x}}_t^u) - \epsilon\right] \leq \gamma\right]$$

$$= \mathbb{P}\left[p_1 = \tilde{v}_t^u, \eta_{\tilde{v}_t^u}(\tilde{\mathbf{x}}_t^u) \geq \eta_{p_2}(\tilde{\mathbf{x}}_t^u), \sum_{h=1}^{H} w_h\left[\sum_{v_j \in \mathcal{V}} \tau_{v_j \tilde{v}_t^u}\eta_{v_j}(\tilde{\mathbf{x}}_t^u) - \epsilon\right] \leq \gamma\right] \tag{3}$$

$$\leq \mathbb{P}\left[p_1 = \tilde{v}_t^u, \eta_{\tilde{v}_t^u}(\tilde{\mathbf{x}}_t^u) \geq \eta_{p_2}(\tilde{\mathbf{x}}_t^u), \tau_{\tilde{v}_t^u \tilde{v}_t^u}(\tilde{\mathbf{x}}_t^u)\eta_{\tilde{v}_t^u}(\tilde{\mathbf{x}}_t^u) + \sum_{v_j \neq \tilde{v}_t^u} \tau_{v_j \tilde{v}_t^u}(\tilde{\mathbf{x}}_t^u)\eta_{v_j}(\tilde{\mathbf{x}}_t^u) \leq \gamma + \epsilon\right]$$

$$= \mathbb{P}\left[p_1 = \tilde{v}_t^u, \eta_{p_2}(\tilde{\mathbf{x}}_t^u) \leq \eta_{\tilde{v}_t^u}(\tilde{\mathbf{x}}_t^u) \leq \frac{\gamma + \epsilon - \sum_{v_j \neq \tilde{v}_t^u} \tau_{v_j \tilde{v}_t^u}(\tilde{\mathbf{x}}_t^u)\eta_{v_j}(\tilde{\mathbf{x}}_t^u)}{\tau_{\tilde{v}_t^u \tilde{v}_t^u}(\tilde{\mathbf{x}}_t^u)}\right]$$

By replacing $\gamma$ with $\tau_{\tilde{v}_t^u \tilde{v}_t^u}(\tilde{\mathbf{x}}_t^u)\eta_{p_2}(\tilde{\mathbf{x}}_t^u) + \sum_{v_j \neq \tilde{v}_t^u} \tau_{v_j \tilde{v}_t^u}(\tilde{\mathbf{x}}_t^u)\eta_{v_j}(\tilde{\mathbf{x}}_t^u)$ , we obtain:

$$\mathbb{P}\left[p_1 = \tilde{v}_t^u, \sum_{h=1}^{H}\left[w_h f_{\tilde{v}_t^u}^h(\tilde{\mathbf{x}}_t^u)\right] \le \gamma\right] \le \mathbb{P}\left[\eta_{p_2}(\tilde{\mathbf{x}}_t^u) \le \eta_{p_1}(\tilde{\mathbf{x}}_t^u) \le \eta_{p_2}(\tilde{\mathbf{x}}_t^u) + \frac{\epsilon}{\tau_{\tilde{v}_t^u \tilde{v}_t^u}(\tilde{\mathbf{x}}_t^u)}\right] \quad (4)$$

Recall that $\epsilon \le \alpha_0 \tau_{\tilde{v}_t^u \tilde{v}_t^u}(\tilde{\mathbf{x}}_t^u)$, which implies $\frac{\epsilon}{\tau_{\tilde{v}_t^u \tilde{v}_t^u}(\tilde{\mathbf{x}}_t^u)} \le \alpha_0$. Hence, the relaxed Multiclass Tsybakov Condition holds and the probability is bounded by $C\left(\frac{\epsilon}{\tau_{\tilde{v}_t^u \tilde{v}_t^u}(\tilde{\mathbf{x}}_t^u)}\right)^\lambda$, namely, $C\left(\mathcal{O}(\epsilon)\right)^\lambda$. $\quad\square$

## B.2 The Proof of Lemma 1

$$\boldsymbol{E}_{\mathrm{DRUT}} = \underbrace{\mathbb{P}\left[p_1 = \tilde{v}_t^u, p_1 \ne v_m, \frac{\sum_{h=1}^{H}\left[w_h f_{\tilde{v}_t^u}^h(\tilde{\mathbf{x}}_t^u)\right]}{\sum_{h=1}^{H}\left[w_h f_{v_m}^h(\tilde{\mathbf{x}}_t^u)\right]} < \beta\right]}_{\text{Case-1}} + \underbrace{\mathbb{P}\left[p_1 \ne \tilde{v}_t^u, p_1 = v_m, \frac{\sum_{h=1}^{H}\left[w_h f_{\tilde{v}_t^u}^h(\tilde{\mathbf{x}}_t^u)\right]}{\sum_{h=1}^{H}\left[w_h f_{v_m}^h(\tilde{\mathbf{x}}_t^u)\right]} \ge \beta\right]}_{\text{Case-2}} + \underbrace{\mathbb{P}\left[p_1 \ne \tilde{v}_t^u, p_1 \ne v_m\right]}_{\text{Case-3}}.$$

**Lemma 1.** *Given Assumption 1 and the set of weights $\left\{w_h \mid 1 \le h \le H, 0 \le w_h \le 1, \sum_{h=1}^{H} w_h = 1\right\}$, $\forall \langle \tilde{\mathbf{x}}_t^u, \tilde{v}_t^u \rangle$, assume $\epsilon \le \min\left[\alpha_0 \tau_{\tilde{v}_t^u \tilde{v}_t^u}(\tilde{\mathbf{x}}_t^u), \alpha_0 \tau_{v_m v_m}(\tilde{\mathbf{x}}_t^u)\right]$. Let $\beta_1 = \left[\frac{\tau_{\tilde{v}_t^u \tilde{v}_t^u}(\tilde{\mathbf{x}}_t^u)\eta_{p_2}(\tilde{\mathbf{x}}_t^u) + \sum_{v_j \ne \tilde{v}_t^u} \tau_{v_j \tilde{v}_t^u} \eta_{v_j}(\tilde{\mathbf{x}}_t^u)}{\sum_{h=1}^{H}\left[w_h f_{v_m}^h(\tilde{\mathbf{x}}_t^u)\right]}\right]$ and $\beta_2 = \left[\frac{\sum_{h=1}^{H}\left[w_h f_{\tilde{v}_t^u}^h(\tilde{\mathbf{x}}_t^u)\right]}{\tau_{v_m v_m}(\tilde{\mathbf{x}}_t^u)\eta_{p_2}(\tilde{\mathbf{x}}_t^u) + \sum_{v_j \ne v_m} \tau_{v_j v_m}(\tilde{\mathbf{x}}_t^u)\eta_{v_j}(\tilde{\mathbf{x}}_t^u)}\right]$. We have: $\beta \le \beta_1$ guarantees the probability of Case-1 in $\boldsymbol{E}_{\mathrm{DRUT}}$ is bounded by $C\left(\mathcal{O}(\epsilon)\right)^\lambda$, and $\beta \ge \beta_2$ guarantees the probability of Case-2 in $\boldsymbol{E}_{\mathrm{DRUT}}$ is bounded by $C\left(\mathcal{O}(\epsilon)\right)^\lambda$.*

*Proof.* For Case-1 of $\boldsymbol{E}_{\mathrm{DRUT}}$, we have:

$$\mathbb{P}\left[p_1 = \tilde{v}_t^u, p_1 \ne v_m, \frac{\sum_{h=1}^{H}\left[w_h f_{\tilde{v}_t^u}^h(\tilde{\mathbf{x}}_t^u)\right]}{\sum_{h=1}^{H}\left[w_h f_{v_m}^h(\tilde{\mathbf{x}}_t^u)\right]} < \beta_1\right]$$

$$\le \mathbb{P}\left[p_1 = \tilde{v}_t^u, \frac{\sum_{h=1}^{H}\left[w_h f_{\tilde{v}_t^u}^h(\tilde{\mathbf{x}}_t^u)\right]}{\sum_{h=1}^{H}\left[w_h f_{v_m}^h(\tilde{\mathbf{x}}_t^u)\right]} < \beta_1\right]$$

$$\le \mathbb{P}\left[p_1 = \tilde{v}_t^u, \eta_{\tilde{v}_t^u}(\tilde{\mathbf{x}}_t^u) \ge \eta_{p_2}(\tilde{\mathbf{x}}_t^u), \sum_{h=1}^{H} w_h\left[\tilde{\eta}_{\tilde{v}_t^u}(\tilde{\mathbf{x}}_t^u) - \epsilon\right] < \beta_1 \sum_{h=1}^{H}\left[w_h f_{v_m}^h(\tilde{\mathbf{x}}_t^u)\right]\right]$$

$$= \mathbb{P}\left[p_1 = \tilde{v}_t^u, \eta_{\tilde{v}_t^u}(\tilde{\mathbf{x}}_t^u) \ge \eta_{p_2}(\tilde{\mathbf{x}}_t^u), \tilde{\eta}_{\tilde{v}_t^u}(\tilde{\mathbf{x}}_t^u) < \beta_1 \sum_{h=1}^{H}\left[w_h f_{v_m}^h(\tilde{\mathbf{x}}_t^u)\right] + \epsilon\right]$$

$$= \mathbb{P}\left[p_1 = \tilde{v}_t^u, \eta_{\tilde{v}_t^u}(\tilde{\mathbf{x}}_t^u) \ge \eta_{p_2}(\tilde{\mathbf{x}}_t^u), \sum_{v_j \in \mathcal{V}} \tau_{v_j \tilde{v}_t^u} \eta_{v_j}(\tilde{\mathbf{x}}_t^u) < \beta_1 \sum_{h=1}^{H}\left[w_h f_{v_m}^h(\tilde{\mathbf{x}}_t^u)\right] + \epsilon\right]$$

$$= \mathbb{P}\left[p_1 = \tilde{v}_t^u, \eta_{p_2}(\tilde{\mathbf{x}}_t^u) \le \eta_{\tilde{v}_t^u}(\tilde{\mathbf{x}}_t^u) < \frac{\beta_1 \sum_{h=1}^{H}\left[w_h f_{v_m}^h(\tilde{\mathbf{x}}_t^u)\right] + \epsilon - \sum_{v_j \ne \tilde{v}_t^u}\left[\tau_{v_j \tilde{v}_t^u} \eta_{v_j}(\tilde{\mathbf{x}}_t^u)\right]}{\tau_{\tilde{v}_t^u \tilde{v}_t^u}(\tilde{\mathbf{x}}_t^u)}\right].$$
(5)

By substituting $\beta_1$ with $\frac{\tau_{\tilde{v}_t^u \tilde{v}_t^u}(\tilde{\mathbf{x}}_t^u)\eta_{p_2}(\tilde{\mathbf{x}}_t^u) + \sum_{v_j \ne \tilde{v}_t^u} \tau_{v_j \tilde{v}_t^u} \eta_{v_j}(\tilde{\mathbf{x}}_t^u)}{\sum_{h=1}^{H}\left[w_h f_{v_m}^h(\tilde{\mathbf{x}}_t^u)\right]}$ in Eq. 7, we obtain:

$$\mathbb{P}\left[p_1 = \tilde{v}_t^u, p_1 \ne v_m, \frac{\sum_{h=1}^{H}\left[w_h f_{\tilde{v}_t^u}^h(\tilde{\mathbf{x}}_t^u)\right]}{\sum_{h=1}^{H}\left[w_h f_{v_m}^h(\tilde{\mathbf{x}}_t^u)\right]} < \beta_1\right]$$
$$\le \mathbb{P}\left[\eta_{p_2}(\tilde{\mathbf{x}}_t^u) \le \eta_{p_1}(\tilde{\mathbf{x}}_t^u) < \eta_{p_2}(\tilde{\mathbf{x}}_t^u) + \frac{\epsilon}{\tau_{\tilde{v}_t^u \tilde{v}_t^u}(\tilde{\mathbf{x}}_t^u)}\right].$$
(6)

Recall that $\epsilon \le \alpha_0 \tau_{\tilde{v}_t^u \tilde{v}_t^u}(\tilde{\mathbf{x}}_t^u)$, which implies $\frac{\epsilon}{\tau_{\tilde{v}_t^u \tilde{v}_t^u}(\tilde{\mathbf{x}}_t^u)} \le \alpha_0$. Hence, the Multiclass Tsybakov Condition holds and the probability of Case-1 is bounded by $C\left(\frac{\epsilon}{\tau_{\tilde{v}_t^u \tilde{v}_t^u}(\tilde{\mathbf{x}}_t^u)}\right)^\lambda$, namely, $C\left(\mathcal{O}(\epsilon)\right)^\lambda$.

Thereafter, for Case-2 of $\boldsymbol{E}_{\text{DRUT}}$, we have:

$$\mathbb{P}\left[p_1 \neq \tilde{v}_t^u, p_1 = v_m, \frac{\sum_{h=1}^H \left[w_h f_{\tilde{v}_t^u}^h(\tilde{\mathbf{x}}_t^u)\right]}{\sum_{h=1}^H \left[w_h f_{v_m}^h(\tilde{\mathbf{x}}_t^u)\right]} \geq \beta_2\right]$$

$$\leq \mathbb{P}\left[p_1 = v_m, \frac{\sum_{h=1}^H \left[w_h f_{\tilde{v}_t^u}^h(\tilde{\mathbf{x}}_t^u)\right]}{\sum_{h=1}^H \left[w_h f_{v_m}^h(\tilde{\mathbf{x}}_t^u)\right]} \geq \beta_2\right]$$

$$\leq \mathbb{P}\left[p_1 = v_m, \eta_m(\tilde{\mathbf{x}}_t^u) \geq \eta_{p_2}(\tilde{\mathbf{x}}_t^u), \sum_{h=1}^H w_h\left[\tilde{\eta}_m(\tilde{\mathbf{x}}_t^u) - \epsilon\right] \leq \frac{\sum_{h=1}^H \left[w_h f_{v_m}^h(\tilde{\mathbf{x}}_t^u)\right]}{\beta_2}\right]$$

$$= \mathbb{P}\left[p_1 = v_m, \eta_m(\tilde{\mathbf{x}}_t^u) \geq \eta_{p_2}(\tilde{\mathbf{x}}_t^u), \tilde{\eta}_m(\tilde{\mathbf{x}}_t^u) \leq \frac{\sum_{h=1}^H \left[w_h f_{v_m}^h(\tilde{\mathbf{x}}_t^u)\right]}{\beta_2} + \epsilon\right]$$

$$= \mathbb{P}\left[p_1 = v_m, \eta_m(\tilde{\mathbf{x}}_t^u) \geq \eta_{p_2}(\tilde{\mathbf{x}}_t^u), \sum_{v_j \in \mathcal{V}} \tau_{v_j v_m}(\tilde{\mathbf{x}}_t^u)\eta_{v_j}(\tilde{\mathbf{x}}_t^u) \leq \frac{\sum_{h=1}^H \left[w_h f_{v_m}^h(\tilde{\mathbf{x}}_t^u)\right]}{\beta_2} + \epsilon\right]$$

$$= \mathbb{P}\left[p_1 = v_m, \eta_{p_2}(\tilde{\mathbf{x}}_t^u) \leq \eta_m(\tilde{\mathbf{x}}_t^u) \leq \frac{\frac{\sum_{h=1}^H [w_h f_{v_m}^h(\tilde{\mathbf{x}}_t^u)]}{\beta_2} - \sum_{v_j \neq v_m} \left[\tau_{v_j v_m}(\tilde{\mathbf{x}}_t^u)\eta_{v_j}(\tilde{\mathbf{x}}_t^u)\right]}{\tau_{v_m v_m}(\tilde{\mathbf{x}}_t^u)} + \frac{\epsilon}{\tau_{v_m v_m}(\tilde{\mathbf{x}}_t^u)}\right].$$

(7)

We replace $\beta_2$ with $\frac{\sum_{h=1}^H [w_h f_{\tilde{v}_t^u}^h(\tilde{\mathbf{x}}_t^u)]}{\tau_{v_m v_m}(\tilde{\mathbf{x}}_t^u)\eta_{p_2}(\tilde{\mathbf{x}}_t^u) + \sum_{v_j \neq v_m} \tau_{v_j v_m}(\tilde{\mathbf{x}}_t^u)\eta_{v_j}(\tilde{\mathbf{x}}_t^u)}$ in Eq. 5 and continue the calculation:

$$\mathbb{P}\left[p_1 \neq \tilde{v}_t^u, p_1 = v_m, \frac{\sum_{h=1}^H \left[w_h f_{\tilde{v}_t^u}^h(\tilde{\mathbf{x}}_t^u)\right]}{\sum_{h=1}^H \left[w_h f_{v_m}^h(\tilde{\mathbf{x}}_t^u)\right]} \geq \beta_2\right]$$

$$\leq \mathbb{P}\left[\eta_{p_2}(\tilde{\mathbf{x}}_t^u) \leq \eta_m(\tilde{\mathbf{x}}_t^u) \leq \eta_{p_2}(\tilde{\mathbf{x}}_t^u) + \frac{\epsilon}{\tau_{v_m v_m}(\tilde{\mathbf{x}}_t^u)}\right].$$

(8)

Recall that $\epsilon \leq \alpha_0 \tau_{v_m v_m}(\tilde{\mathbf{x}}_t^u)$, which implies $\frac{\epsilon}{\tau_{v_m v_m}(\tilde{\mathbf{x}}_t^u)} \leq \alpha_0$. Hence, the Multiclass Tsybakov Condition holds and the probabiliy of Case-2 is bounded by $C\left(\frac{\epsilon}{\tau_{v_m v_m}(\tilde{\mathbf{x}}_t^u)}\right)^\lambda$, namely, $C\left(\mathcal{O}(\epsilon)\right)^\lambda$. $\quad\square$

### B.3 The Proof of Theorem 2

**Theorem 2** (The Upper Bound of $\boldsymbol{E}_{\text{DRUT}}$). *Given Assumption 1 and the set of weights $\{w_h \mid 1 \leq h \leq H, 0 \leq w_h \leq 1, \sum_{h=1}^H w_h = 1\}$, $\forall \langle \tilde{\mathbf{x}}_t^u, \tilde{v}_t^u \rangle$, let $\beta_1 = \left[\frac{\tau_{\tilde{v}_t^u \tilde{v}_t^u}(\tilde{\mathbf{x}}_t^u)\eta_{p_2}(\tilde{\mathbf{x}}_t^u) + \sum_{v_j \neq \tilde{v}_t^u} \tau_{v_j \tilde{v}_t^u}\eta_{v_j}(\tilde{\mathbf{x}}_t^u)}{\sum_{h=1}^H [w_h f_{v_m}^h(\tilde{\mathbf{x}}_t^u)]}\right]$, $\beta_2 = \left[\frac{\sum_{h=1}^H [w_h f_{\tilde{v}_t^u}^h(\tilde{\mathbf{x}}_t^u)]}{\tau_{v_m v_m}(\tilde{\mathbf{x}}_t^u)\eta_{p_2}(\tilde{\mathbf{x}}_t^u) + \sum_{v_j \neq v_m} \tau_{v_j v_m}(\tilde{\mathbf{x}}_t^u)\eta_{v_j}(\tilde{\mathbf{x}}_t^u)}\right]$, $\xi_1 = |\beta - \beta_1|$, and $\xi_2 = |\beta - \beta_2|$. Assume $\xi_2 < \beta_2$, $\epsilon \leq \min\left[\alpha_0 \tau_{\tilde{v}_t^u \tilde{v}_t^u}(\tilde{\mathbf{x}}_t^u) - \xi_1, \frac{\alpha_0 \tau_{v_m v_m}(\tilde{\mathbf{x}}_t^u)\beta_2(\beta_2 - \xi_2) - \xi_2}{\beta_2(\beta_2 - \xi_2)}, \frac{1}{2}\left[[\tau_{p_1 p_1}(\tilde{\mathbf{x}}_t^u) - \tau_{p_1 v_m}(\tilde{\mathbf{x}}_t^u)][\alpha_0 + \eta_{p_2}(\tilde{\mathbf{x}}_t^u)] - \sum_{v_j \neq p_1} [\tau_{v_j v_m}(\tilde{\mathbf{x}}_t^u) - \tau_{v_j p_1}(\tilde{\mathbf{x}}_t^u)]\eta_{v_j}(\tilde{\mathbf{x}}_t^u)\right]\right]$. We have: $\boldsymbol{E}_{\text{DRUT}} \leq C\left(\mathcal{O}(\epsilon + \xi_1)\right)^\lambda + C\left(\mathcal{O}(\epsilon + \xi_2)\right)^\lambda + C\left(\mathcal{O}(\epsilon)\right)^\lambda$.*

*Proof.* For Case-1 of $\boldsymbol{E}_{\mathrm{DRUT}}$, we have:

$$\mathbb{P}\left[p_1 = \tilde{v}_t^u, p_1 \neq v_m, \frac{\sum_{h=1}^H \left[w_h f_{\tilde{v}_t^u}^h(\tilde{\mathbf{x}}_t^u)\right]}{\sum_{h=1}^H \left[w_h f_{v_m}^h(\tilde{\mathbf{x}}_t^u)\right]} < \beta\right]$$

$$\leq\mathbb{P}\left[p_1 = \tilde{v}_t^u, \frac{\sum_{h=1}^H \left[w_h f_{\tilde{v}_t^u}^h(\tilde{\mathbf{x}}_t^u)\right]}{\sum_{h=1}^H \left[w_h f_{v_m}^h(\tilde{\mathbf{x}}_t^u)\right]} < \beta\right]$$

$$\leq\mathbb{P}\left[p_1 = \tilde{v}_t^u, \eta_{\tilde{v}_t^u}(\tilde{\mathbf{x}}_t^u) \geq \eta_{p_2}(\tilde{\mathbf{x}}_t^u), \sum_{h=1}^H w_h\left[\tilde{\eta}_{\tilde{v}_t^u}(\tilde{\mathbf{x}}_t^u) - \epsilon\right] < \beta\sum_{h=1}^H \left[w_h f_{v_m}^h(\tilde{\mathbf{x}}_t^u)\right]\right]$$

$$=\mathbb{P}\left[p_1 = \tilde{v}_t^u, \eta_{\tilde{v}_t^u}(\tilde{\mathbf{x}}_t^u) \geq \eta_{p_2}(\tilde{\mathbf{x}}_t^u), \tilde{\eta}_{\tilde{v}_t^u}(\tilde{\mathbf{x}}_t^u) < \beta\sum_{h=1}^H \left[w_h f_{v_m}^h(\tilde{\mathbf{x}}_t^u)\right] + \epsilon\right]$$

$$=\mathbb{P}\left[p_1 = \tilde{v}_t^u, \eta_{\tilde{v}_t^u}(\tilde{\mathbf{x}}_t^u) \geq \eta_{p_2}(\tilde{\mathbf{x}}_t^u), \sum_{v_j \in \mathcal{V}} \tau_{v_j \tilde{v}_t^u} \eta_{v_j}(\tilde{\mathbf{x}}_t^u) < \beta\sum_{h=1}^H \left[w_h f_{v_m}^h(\tilde{\mathbf{x}}_t^u)\right] + \epsilon\right]$$

$$\leq\mathbb{P}\left[p_1 = \tilde{v}_t^u, \eta_{p_2}(\tilde{\mathbf{x}}_t^u) \leq \eta_{\tilde{v}_t^u}(\tilde{\mathbf{x}}_t^u) < \frac{\beta\sum_{h=1}^H \left[w_h f_{v_m}^h(\tilde{\mathbf{x}}_t^u)\right] + \epsilon - \sum_{v_j \neq \tilde{v}_t^u} \left[\tau_{v_j \tilde{v}_t^u} \eta_{v_j}(\tilde{\mathbf{x}}_t^u)\right]}{\tau_{\tilde{v}_t^u \tilde{v}_t^u}(\tilde{\mathbf{x}}_t^u)}\right]$$

$$\leq\mathbb{P}\left[p_1 = \tilde{v}_t^u, \eta_{p_2}(\tilde{\mathbf{x}}_t^u) \leq \eta_{\tilde{v}_t^u}(\tilde{\mathbf{x}}_t^u) < \frac{(\beta_1 + \xi_1)\sum_{h=1}^H \left[w_h f_{v_m}^h(\tilde{\mathbf{x}}_t^u)\right] + \epsilon - \sum_{v_j \neq \tilde{v}_t^u} \left[\tau_{v_j \tilde{v}_t^u} \eta_{v_j}(\tilde{\mathbf{x}}_t^u)\right]}{\tau_{\tilde{v}_t^u \tilde{v}_t^u}(\tilde{\mathbf{x}}_t^u)}\right].$$

$$(9)$$

We substitute $\beta_1$ with $\frac{\tau_{\tilde{v}_t^u \tilde{v}_t^u}(\tilde{\mathbf{x}}_t^u)\eta_{p_2}(\tilde{\mathbf{x}}_t^u) + \sum_{v_j \neq \tilde{v}_t^u} \tau_{v_j \tilde{v}_t^u} \eta_{v_j}(\tilde{\mathbf{x}}_t^u)}{\sum_{h=1}^H [w_h f_{v_m}^h(\tilde{\mathbf{x}}_t^u)]}$ and obtain:

$$\mathbb{P}\left[p_1 = \tilde{v}_t^u, p_1 \neq v_m, \frac{\sum_{h=1}^H \left[w_h f_{\tilde{v}_t^u}^h(\tilde{\mathbf{x}}_t^u)\right]}{\sum_{h=1}^H \left[w_h f_{v_m}^h(\tilde{\mathbf{x}}_t^u)\right]} < \beta\right]$$

$$\leq\mathbb{P}\left[p_1 = \tilde{v}_t^u, \eta_{p_2}(\tilde{\mathbf{x}}_t^u) \leq \eta_{\tilde{v}_t^u}(\tilde{\mathbf{x}}_t^u) < \eta_{p_2}(\tilde{\mathbf{x}}_t^u) + \frac{\epsilon}{\tau_{\tilde{v}_t^u \tilde{v}_t^u}(\tilde{\mathbf{x}}_t^u)} + \frac{\xi_1 \sum_{h=1}^H \left[w_h f_{v_m}^h(\tilde{\mathbf{x}}_t^u)\right]}{\tau_{\tilde{v}_t^u \tilde{v}_t^u}(\tilde{\mathbf{x}}_t^u)}\right] \quad (10)$$

$$\leq\mathbb{P}\left[\eta_{p_2}(\tilde{\mathbf{x}}_t^u) \leq \eta_{p_1}(\tilde{\mathbf{x}}_t^u) < \eta_{p_2}(\tilde{\mathbf{x}}_t^u) + \frac{\epsilon + \xi_1}{\tau_{\tilde{v}_t^u \tilde{v}_t^u}(\tilde{\mathbf{x}}_t^u)}\right].$$

Recall that $\epsilon \leq \alpha_0 \tau_{\tilde{v}_t^u \tilde{v}_t^u}(\tilde{\mathbf{x}}_t^u) - \xi_1$, which implies $\frac{\epsilon + \xi_1}{\tau_{\tilde{v}_t^u \tilde{v}_t^u}(\tilde{\mathbf{x}}_t^u)} \leq \alpha_0$. Hence, the Multiclass Tsybakov Condition holds and the probabiliy of Case-2 is bounded by $C\left(\frac{\epsilon + \xi_1}{\tau_{\tilde{v}_t^u \tilde{v}_t^u}(\tilde{\mathbf{x}}_t^u)}\right)^\lambda$, namely, $C\left(\mathcal{O}(\epsilon + \xi_1)\right)^\lambda$.

For Case-2 of $\boldsymbol{E}_{\mathrm{DRUT}}$, we have:

$$
\mathbb{P}\left[p_1 \neq \tilde{v}_t^u, p_1 = v_m, \frac{\sum_{h=1}^{H}\left[w_h f_{\tilde{v}_t^u}^h(\tilde{\mathbf{x}}_t^u)\right]}{\sum_{h=1}^{H}\left[w_h f_{v_m}^h(\tilde{\mathbf{x}}_t^u)\right]} \geq \beta\right]
$$

$$
\leq \mathbb{P}\left[p_1 = v_m, \frac{\sum_{h=1}^{H}\left[w_h f_{\tilde{v}_t^u}^h(\tilde{\mathbf{x}}_t^u)\right]}{\sum_{h=1}^{H}\left[w_h f_{v_m}^h(\tilde{\mathbf{x}}_t^u)\right]} \geq \beta\right]
$$

$$
\leq \mathbb{P}\left[p_1 = v_m, \eta_m(\tilde{\mathbf{x}}_t^u) \geq \eta_{p_2}(\tilde{\mathbf{x}}_t^u), \sum_{h=1}^{H} w_h\left[\tilde{\eta}_m(\tilde{\mathbf{x}}_t^u) - \epsilon\right] \leq \frac{\sum_{h=1}^{H}\left[w_h f_{v_m}^h(\tilde{\mathbf{x}}_t^u)\right]}{\beta}\right]
$$

$$
= \mathbb{P}\left[p_1 = v_m, \eta_m(\tilde{\mathbf{x}}_t^u) \geq \eta_{p_2}(\tilde{\mathbf{x}}_t^u), \tilde{\eta}_m(\tilde{\mathbf{x}}_t^u) \leq \frac{\sum_{h=1}^{H}\left[w_h f_{v_m}^h(\tilde{\mathbf{x}}_t^u)\right]}{\beta} + \epsilon\right]
$$

$$
= \mathbb{P}\left[p_1 = v_m, \eta_m(\tilde{\mathbf{x}}_t^u) \geq \eta_{p_2}(\tilde{\mathbf{x}}_t^u), \sum_{v_j \in \mathcal{V}} \tau_{v_j v_m}(\tilde{\mathbf{x}}_t^u)\eta_{v_j}(\tilde{\mathbf{x}}_t^u) \leq \frac{\sum_{h=1}^{H}\left[w_h f_{v_m}^h(\tilde{\mathbf{x}}_t^u)\right]}{\beta} + \epsilon\right]
$$

$$
\leq \mathbb{P}\left[p_1 = v_m, \eta_{p_2}(\tilde{\mathbf{x}}_t^u) \leq \eta_m(\tilde{\mathbf{x}}_t^u) \leq \frac{\frac{\sum_{h=1}^{H}[w_h f_{v_m}^h(\tilde{\mathbf{x}}_t^u)]}{\beta} - \sum_{v_j \neq v_m}\left[\tau_{v_j v_m}(\tilde{\mathbf{x}}_t^u)\eta_{v_j}(\tilde{\mathbf{x}}_t^u)\right]}{\tau_{v_m v_m}(\tilde{\mathbf{x}}_t^u)} + \frac{\epsilon}{\tau_{v_m v_m}(\tilde{\mathbf{x}}_t^u)}\right]
$$

$$
\leq \mathbb{P}\left[p_1 = v_m, \eta_{p_2}(\tilde{\mathbf{x}}_t^u) \leq \eta_m(\tilde{\mathbf{x}}_t^u) \leq \frac{\frac{\sum_{h=1}^{H}[w_h f_{v_m}^h(\tilde{\mathbf{x}}_t^u)]}{\beta_2 - \xi_2} - \sum_{v_j \neq v_m}\left[\tau_{v_j v_m}(\tilde{\mathbf{x}}_t^u)\eta_{v_j}(\tilde{\mathbf{x}}_t^u)\right]}{\tau_{v_m v_m}(\tilde{\mathbf{x}}_t^u)} + \frac{\epsilon}{\tau_{v_m v_m}(\tilde{\mathbf{x}}_t^u)}\right]
$$

$$
= \mathbb{P}\left[p_1 = v_m, \eta_{p_2}(\tilde{\mathbf{x}}_t^u) \leq \eta_m(\tilde{\mathbf{x}}_t^u) \leq \frac{\frac{\sum_{h=1}^{H}[w_h f_{v_m}^h(\tilde{\mathbf{x}}_t^u)]}{\beta_2} - \sum_{v_j \neq v_m}\left[\tau_{v_j v_m}(\tilde{\mathbf{x}}_t^u)\eta_{v_j}(\tilde{\mathbf{x}}_t^u)\right]}{\tau_{v_m v_m}(\tilde{\mathbf{x}}_t^u)} + \frac{\epsilon}{\tau_{v_m v_m}(\tilde{\mathbf{x}}_t^u)}\right.
$$
$$
\left. + \frac{\xi_2 \sum_{h=1}^{H}[w_h f_{v_m}^h(\tilde{\mathbf{x}}_t^u)]}{\beta_2(\beta_2 - \xi_2)\tau_{v_m v_m}(\tilde{\mathbf{x}}_t^u)}\right]
$$

(11)

We replace $\beta_2$ with $\frac{\sum_{h=1}^{H}[w_h f_{\tilde{v}_t^u}^h(\tilde{\mathbf{x}}_t^u)]}{\tau_{v_m v_m}(\tilde{\mathbf{x}}_t^u)\eta_{p_2}(\tilde{\mathbf{x}}_t^u) + \sum_{v_j \neq v_m} \tau_{v_j v_m}(\tilde{\mathbf{x}}_t^u)\eta_{v_j}(\tilde{\mathbf{x}}_t^u)}$ and continue the calculation:

$$
\mathbb{P}\left[p_1 \neq \tilde{v}_t^u, p_1 = v_m, \frac{\sum_{h=1}^{H}\left[w_h f_{\tilde{v}_t^u}^h(\tilde{\mathbf{x}}_t^u)\right]}{\sum_{h=1}^{H}\left[w_h f_{v_m}^h(\tilde{\mathbf{x}}_t^u)\right]} \geq \beta\right]
$$

$$
\leq \mathbb{P}\left[p_1 = v_m, \eta_{p_2}(\tilde{\mathbf{x}}_t^u) \leq \eta_m(\tilde{\mathbf{x}}_t^u) \leq \eta_{p_2}(\tilde{\mathbf{x}}_t^u) + \frac{\epsilon}{\tau_{v_m v_m}(\tilde{\mathbf{x}}_t^u)} + \frac{\xi_2 \sum_{h=1}^{H}[w_h f_{v_m}^h(\tilde{\mathbf{x}}_t^u)]}{\beta_2(\beta_2 - \xi_2)\tau_{v_m v_m}(\tilde{\mathbf{x}}_t^u)}\right] \quad (12)
$$

$$
\leq \mathbb{P}\left[\eta_{p_2}(\tilde{\mathbf{x}}_t^u) \leq \eta_{p_1}(\tilde{\mathbf{x}}_t^u) \leq \eta_{p_2}(\tilde{\mathbf{x}}_t^u) + \frac{\epsilon}{\tau_{v_m v_m}(\tilde{\mathbf{x}}_t^u)} + \frac{\xi_2}{\beta_2(\beta_2 - \xi_2)\tau_{v_m v_m}(\tilde{\mathbf{x}}_t^u)}\right].
$$

Recall that $\epsilon \leq \frac{\alpha_0 \tau_{v_m v_m}(\tilde{\mathbf{x}}_t^u)\beta_2(\beta_2 - \xi_2) - \xi_2}{\beta_2(\beta_2 - \xi_2)}$, which implies $\frac{\epsilon}{\tau_{v_m v_m}(\tilde{\mathbf{x}}_t^u)} + \frac{\xi_2}{\beta_2(\beta_2 - \xi_2)\tau_{v_m v_m}(\tilde{\mathbf{x}}_t^u)} \leq \alpha_0$.
Hence, the relaxed Multiclass Tsybakov Condition holds and the probability of Case-2 is bounded by
$C\left(\frac{\epsilon\beta_2(\beta_2 - \xi_2) + \xi_2}{\beta_2(\beta_2 - \xi_2)\tau_{v_m v_m}(\tilde{\mathbf{x}}_t^u)}\right)^{\lambda}$, namely, $C\left(\mathcal{O}(\epsilon + \xi_2)\right)^{\lambda}$.

Finally, for Case-3 of $\boldsymbol{E}_{\mathrm{DRUT}}$, we have:

$$
\mathbb{P}\Big[p_1 \neq \tilde{v}_t^u, p_1 \neq v_m\Big]
$$

$$
\leq \mathbb{P}\Big[p_1 \neq v_m\Big]
$$

$$
= \mathbb{P}\Big[p_1 \neq v_m, \eta_{p_1}(\tilde{\mathbf{x}}_t^u) \geq \eta_{p_2}(\tilde{\mathbf{x}}_t^u), \frac{\sum_{h=1}^{H}\big[w_h f_{p_1}^h(\tilde{\mathbf{x}}_t^u)\big]}{\sum_{h=1}^{H}\big[w_h f_{v_m}^h(\tilde{\mathbf{x}}_t^u)\big]} < 1\Big]
$$

$$
\leq \mathbb{P}\Big[\eta_{p_1}(\tilde{\mathbf{x}}_t^u) \geq \eta_{p_2}(\tilde{\mathbf{x}}_t^u), \sum_{h=1}^{H}\big[w_h f_{p_1}^h(\tilde{\mathbf{x}}_t^u)\big] < \sum_{h=1}^{H}\big[w_h f_{v_m}^h(\tilde{\mathbf{x}}_t^u)\big]\Big]
$$

$$
\leq \mathbb{P}\Big[\eta_{p_1}(\tilde{\mathbf{x}}_t^u) \geq \eta_{p_2}(\tilde{\mathbf{x}}_t^u), \sum_{h=1}^{H} w_h\big[\tilde{\eta}_{p_1}(\tilde{\mathbf{x}}_t^u) - \epsilon\big] < \sum_{h=1}^{H} w_h\big[\tilde{\eta}_m(\tilde{\mathbf{x}}_t^u) + \epsilon\big]\Big]
$$

$$
= \mathbb{P}\Big[\eta_{p_1}(\tilde{\mathbf{x}}_t^u) \geq \eta_{p_2}(\tilde{\mathbf{x}}_t^u), \tilde{\eta}_{p_1}(\tilde{\mathbf{x}}_t^u) - \epsilon < \tilde{\eta}_m(\tilde{\mathbf{x}}_t^u) + \epsilon\Big]
$$

$$
= \mathbb{P}\Big[\eta_{p_1}(\tilde{\mathbf{x}}_t^u) \geq \eta_{p_2}(\tilde{\mathbf{x}}_t^u), \sum_{v_j \in \mathcal{V}} \tau_{v_j p_1}(\tilde{\mathbf{x}}_t^u)\eta_{v_j}(\tilde{\mathbf{x}}_t^u) < \sum_{v_j \in \mathcal{V}} \tau_{v_j v_m}(\tilde{\mathbf{x}}_t^u)\eta_{v_j}(\tilde{\mathbf{x}}_t^u) + 2\epsilon\Big]
$$

$$
= \mathbb{P}\Big[\eta_{p_1}(\tilde{\mathbf{x}}_t^u) \geq \eta_{p_2}(\tilde{\mathbf{x}}_t^u), \tau_{p_1 p_1}(\tilde{\mathbf{x}}_t^u)\eta_{p_1}(\tilde{\mathbf{x}}_t^u) + \sum_{v_j \neq p_1} \tau_{v_j p_1}(\tilde{\mathbf{x}}_t^u)\eta_{v_j}(\tilde{\mathbf{x}}_t^u) < \tag{13}
$$
$$
\tau_{p_1 v_m}(\tilde{\mathbf{x}}_t^u)\eta_{p_1}(\tilde{\mathbf{x}}_t^u) + \sum_{v_j \neq p_1} \tau_{v_j v_m}(\tilde{\mathbf{x}}_t^u)\eta_{v_j}(\tilde{\mathbf{x}}_t^u) + 2\epsilon\Big]
$$

$$
= \mathbb{P}\Big[\eta_{p_1}(\tilde{\mathbf{x}}_t^u) \geq \eta_{p_2}(\tilde{\mathbf{x}}_t^u), \eta_{p_1}(\tilde{\mathbf{x}}_t^u)\big[\tau_{p_1 p_1}(\tilde{\mathbf{x}}_t^u) - \tau_{p_1 v_m}(\tilde{\mathbf{x}}_t^u)\big] <
$$
$$
\sum_{v_j \neq p_1}\big[\tau_{v_j v_m}(\tilde{\mathbf{x}}_t^u) - \tau_{v_j p_1}(\tilde{\mathbf{x}}_t^u)\big]\eta_{v_j}(\tilde{\mathbf{x}}_t^u) + 2\epsilon\Big]
$$

$$
= \mathbb{P}\Big[\eta_{p_1}(\tilde{\mathbf{x}}_t^u) \geq \eta_{p_2}(\tilde{\mathbf{x}}_t^u), \eta_{p_1}(\tilde{\mathbf{x}}_t^u) < \frac{\sum_{v_j \neq p_1}\big[\tau_{v_j v_m}(\tilde{\mathbf{x}}_t^u) - \tau_{v_j p_1}(\tilde{\mathbf{x}}_t^u)\big]\eta_{v_j}(\tilde{\mathbf{x}}_t^u) + 2\epsilon}{\big[\tau_{p_1 p_1}(\tilde{\mathbf{x}}_t^u) - \tau_{p_1 v_m}(\tilde{\mathbf{x}}_t^u)\big]}\Big]
$$

$$
= \mathbb{P}\Big[\eta_{p_2}(\tilde{\mathbf{x}}_t^u) \leq \eta_{p_1}(\tilde{\mathbf{x}}_t^u) < \eta_{p_2}(\tilde{\mathbf{x}}_t^u) +
$$
$$
\frac{\sum_{v_j \neq p_1}\big[\tau_{v_j v_m}(\tilde{\mathbf{x}}_t^u) - \tau_{v_j p_1}(\tilde{\mathbf{x}}_t^u)\big]\eta_{v_j}(\tilde{\mathbf{x}}_t^u) + 2\epsilon - \big[\tau_{p_1 p_1}(\tilde{\mathbf{x}}_t^u) - \tau_{p_1 v_m}(\tilde{\mathbf{x}}_t^u)\big]\eta_{p_2}(\tilde{\mathbf{x}}_t^u)}{\big[\tau_{p_1 p_1}(\tilde{\mathbf{x}}_t^u) - \tau_{p_1 v_m}(\tilde{\mathbf{x}}_t^u)\big]}\Big]
$$

Recall that $\epsilon \leq \frac{1}{2}\Big[\big[\tau_{p_1 p_1}(\tilde{\mathbf{x}}_t^u) - \tau_{p_1 v_m}(\tilde{\mathbf{x}}_t^u)\big]\big[\alpha_0 + \eta_{p_2}(\tilde{\mathbf{x}}_t^u)\big] - \sum_{v_j \neq p_1}\big[\tau_{v_j v_m}(\tilde{\mathbf{x}}_t^u) - \tau_{v_j p_1}(\tilde{\mathbf{x}}_t^u)\big]\eta_{v_j}(\tilde{\mathbf{x}}_t^u)\Big]$, which implies $\frac{\sum_{v_j \neq p_1}\big[\tau_{v_j v_m}(\tilde{\mathbf{x}}_t^u) - \tau_{v_j p_1}(\tilde{\mathbf{x}}_t^u)\big]\eta_{v_j}(\tilde{\mathbf{x}}_t^u) + 2\epsilon - \big[\tau_{p_1 p_1}(\tilde{\mathbf{x}}_t^u) - \tau_{p_1 v_m}(\tilde{\mathbf{x}}_t^u)\big]\eta_{p_2}(\tilde{\mathbf{x}}_t^u)}{\big[\tau_{p_1 p_1}(\tilde{\mathbf{x}}_t^u) - \tau_{p_1 v_m}(\tilde{\mathbf{x}}_t^u)\big]} \leq \alpha_0$. Hence, the relaxed Multiclass Tsybakov Condition holds and the probability of Case-3 is bounded by $C\Big(\frac{\sum_{v_j \neq p_1}\big[\tau_{v_j v_m}(\tilde{\mathbf{x}}_t^u) - \tau_{v_j p_1}(\tilde{\mathbf{x}}_t^u)\big]\eta_{v_j}(\tilde{\mathbf{x}}_t^u) + 2\epsilon - \big[\tau_{p_1 p_1}(\tilde{\mathbf{x}}_t^u) - \tau_{p_1 v_m}(\tilde{\mathbf{x}}_t^u)\big]\eta_{p_2}(\tilde{\mathbf{x}}_t^u)}{\big[\tau_{p_1 p_1}(\tilde{\mathbf{x}}_t^u) - \tau_{p_1 v_m}(\tilde{\mathbf{x}}_t^u)\big]}\Big)^{\lambda}$, namely, $C\big(\mathcal{O}(\epsilon)\big)^{\lambda}$.

$\square$

## B.4 The Proof of Theorem 3

$$
\boldsymbol{E}_{\mathrm{DDUI}} = \mathbb{P}\Big[p_1' = \tilde{v}_{t-l}^u, \frac{\sum_{h=1}^{H}\big[w_h \breve{f}_{\tilde{v}_{t-l}^u}^h(\breve{\mathbf{x}}_t^u)\big]}{\sum_{h=1}^{H}\big[w_h \breve{f}_{\bar{v}_m}^h(\breve{\mathbf{x}}_t^u)\big]} < \beta'\Big] + \mathbb{P}\Big[p_1' \neq \tilde{v}_{t-l}^u, \frac{\sum_{h=1}^{H}\big[w_h \breve{f}_{\tilde{v}_{t-l}^u}^h(\breve{\mathbf{x}}_t^u)\big]}{\sum_{h=1}^{H}\big[w_h \breve{f}_{\bar{v}_m}^h(\breve{\mathbf{x}}_t^u)\big]} \geq \beta'\Big]. \tag{14}
$$

**Theorem 3** (The Upper Bound of $\boldsymbol{E}_{\mathrm{DDUI}}$). *Given Assumption 1 and the set of weights* $\{w_h \mid 1 \leq h \leq H, 0 \leq w_h \leq 1, \sum_{h=1}^H w_h = 1\}$, $\forall \langle \bar{\mathbf{x}}_t^u, \tilde{v}_{t-l}^u \rangle$, *let* $\beta_1' = \left[ \frac{\tau'_{\tilde{v}_{t-l}^u \tilde{v}_{t-l}^u}(\bar{\mathbf{x}}_t^u) \eta'_{p_2'}(\bar{\mathbf{x}}_t^u) + \sum_{v_j \neq \tilde{v}_{t-l}^u} \tau'_{v_j \tilde{v}_{t-l}^u} \eta'_{v_j}(\bar{\mathbf{x}}_t^u)}{\sum_{h=1}^H [w_h \bar{f}_{\tilde{v}_m}^h(\bar{\mathbf{x}}_t^u)]} \right]$,

$\beta_2' = \left[ \frac{\sum_{h=1}^H [w_h \bar{f}_{\tilde{v}_{t-l}^u}^h(\bar{\mathbf{x}}_t^u)]}{\tau'_{\tilde{v}_m \tilde{v}_m}(\bar{\mathbf{x}}_t^u) \eta'_{p_2'}(\bar{\mathbf{x}}_t^u) + \sum_{v_j \neq \tilde{v}_m} \tau'_{v_j \tilde{v}_m}(\bar{\mathbf{x}}_t^u) \eta'_{v_j}(\bar{\mathbf{x}}_t^u)} \right]$, $\xi_1' = |\beta' - \beta_1'|$, $\xi_2' = |\beta' - \beta_2'|$. *Assume* $\xi_2' < \beta_2'$,

$\epsilon' \leq \min\left[ \alpha_0 \tau'_{\tilde{v}_{t-l}^u \tilde{v}_{t-l}^u}(\bar{\mathbf{x}}_t^u) - \xi_1', \frac{\alpha_0 \tau'_{\tilde{v}_m \tilde{v}_m}(\bar{\mathbf{x}}_t^u) \beta_2'(\beta_2' - \xi_2') - \xi_2'}{\beta_2'(\beta_2' - \xi_2')}, \frac{1}{2} [[\tau'_{p_1' p_1'}(\bar{\mathbf{x}}_t^u) - \tau'_{p_1' \tilde{v}_m}(\bar{\mathbf{x}}_t^u)][\alpha_0 + \eta'_{p_2'}(\bar{\mathbf{x}}_t^u)] - \sum_{v_j \neq p_1'} [\tau'_{v_j \tilde{v}_m}(\bar{\mathbf{x}}_t^u) - \tau'_{v_j p_1'}(\bar{\mathbf{x}}_t^u)] \eta'_{v_j}(\bar{\mathbf{x}}_t^u)] \right]$. *We have:* $\boldsymbol{E}_{\mathrm{DDUI}} \leq C\big(\mathcal{O}(\epsilon' + \xi_1')\big)^\lambda + C\big(\mathcal{O}(\epsilon' + \xi_2')\big)^\lambda + C\big(\mathcal{O}(\epsilon')\big)^\lambda$.

*Proof.* For the first term of $\boldsymbol{E}_{\mathrm{DDUI}}$, we have:

$$\mathbb{P}\left[ p_1' = \tilde{v}_{t-l}^u, \frac{\sum_{h=1}^H [w_h \bar{f}_{\tilde{v}_{t-l}^u}^h(\bar{\mathbf{x}}_t^u)]}{\sum_{h=1}^H [w_h \bar{f}_{\tilde{v}_m}^h(\bar{\mathbf{x}}_t^u)]} < \beta' \right]$$

$$\leq \mathbb{P}\left[ p_1' = \tilde{v}_{t-l}^u, \eta'_{\tilde{v}_{t-l}^u}(\bar{\mathbf{x}}_t^u) \geq \eta'_{p_2'}(\bar{\mathbf{x}}_t^u), \sum_{h=1}^H w_h [\tilde{\eta}'_{\tilde{v}_{t-l}^u}(\bar{\mathbf{x}}_t^u) - \epsilon'] < \beta' \sum_{h=1}^H [w_h \bar{f}_{v_m}^h(\bar{\mathbf{x}}_t^u)] \right]$$

$$= \mathbb{P}\left[ p_1' = \tilde{v}_{t-l}^u, \eta'_{\tilde{v}_{t-l}^u}(\bar{\mathbf{x}}_t^u) \geq \eta'_{p_2'}(\bar{\mathbf{x}}_t^u), \tilde{\eta}'_{\tilde{v}_{t-l}^u}(\bar{\mathbf{x}}_t^u) < \beta' \sum_{h=1}^H [w_h \bar{f}_{v_m}^h(\bar{\mathbf{x}}_t^u)] + \epsilon' \right]$$

$$= \mathbb{P}\left[ p_1' = \tilde{v}_{t-l}^u, \eta'_{\tilde{v}_{t-l}^u}(\bar{\mathbf{x}}_t^u) \geq \eta'_{p_2'}(\bar{\mathbf{x}}_t^u), \sum_{v_j \in \mathcal{V}} \tau'_{v_j \tilde{v}_{t-l}^u} \eta'_{v_j}(\bar{\mathbf{x}}_t^u) < \beta' \sum_{h=1}^H [w_h \bar{f}_{v_m}^h(\bar{\mathbf{x}}_t^u)] + \epsilon' \right]$$

$$\leq \mathbb{P}\left[ p_1' = \tilde{v}_{t-l}^u, \eta'_{p_2'}(\bar{\mathbf{x}}_t^u) \leq \eta'_{\tilde{v}_{t-l}^u}(\bar{\mathbf{x}}_t^u) < \frac{\beta' \sum_{h=1}^H [w_h \bar{f}_{v_m}^h(\bar{\mathbf{x}}_t^u)] + \epsilon' - \sum_{v_j \neq \tilde{v}_{t-l}^u} [\tau'_{v_j \tilde{v}_{t-l}^u} \eta'_{v_j}(\bar{\mathbf{x}}_t^u)]}{\tau'_{\tilde{v}_{t-l}^u \tilde{v}_{t-l}^u}(\bar{\mathbf{x}}_t^u)} \right]$$

$$\leq \mathbb{P}\left[ p_1' = \tilde{v}_{t-l}^u, \eta'_{p_2'}(\bar{\mathbf{x}}_t^u) \leq \eta'_{\tilde{v}_{t-l}^u}(\bar{\mathbf{x}}_t^u) < \frac{(\beta_1' + \xi_1') \sum_{h=1}^H [w_h \bar{f}_{v_m}^h(\bar{\mathbf{x}}_t^u)] + \epsilon' - \sum_{v_j \neq \tilde{v}_{t-l}^u} [\tau'_{v_j \tilde{v}_{t-l}^u} \eta'_{v_j}(\bar{\mathbf{x}}_t^u)]}{\tau'_{\tilde{v}_{t-l}^u \tilde{v}_{t-l}^u}(\bar{\mathbf{x}}_t^u)} \right].$$

$$(15)$$

We substitute $\beta_1'$ with $\frac{\tau'_{\tilde{v}_{t-l}^u \tilde{v}_{t-l}^u}(\bar{\mathbf{x}}_t^u) \eta'_{p_2'}(\bar{\mathbf{x}}_t^u) + \sum_{v_j \neq \tilde{v}_{t-l}^u} \tau'_{v_j \tilde{v}_{t-l}^u} \eta'_{v_j}(\bar{\mathbf{x}}_t^u)}{\sum_{h=1}^H [w_h \bar{f}_{\tilde{v}_m}^h(\bar{\mathbf{x}}_t^u)]}$ and obtain:

$$\mathbb{P}\left[ p_1' = \tilde{v}_{t-l}^u, \frac{\sum_{h=1}^H [w_h \bar{f}_{\tilde{v}_{t-l}^u}^h(\bar{\mathbf{x}}_t^u)]}{\sum_{h=1}^H [w_h \bar{f}_{\tilde{v}_m}^h(\bar{\mathbf{x}}_t^u)]} < \beta' \right]$$

$$\leq \mathbb{P}\left[ p_1' = \tilde{v}_{t-l}^u, \eta'_{p_2'}(\bar{\mathbf{x}}_t^u) \leq \eta'_{\tilde{v}_{t-l}^u}(\bar{\mathbf{x}}_t^u) < \eta'_{p_2'}(\bar{\mathbf{x}}_t^u) + \frac{\epsilon'}{\tau'_{\tilde{v}_{t-l}^u \tilde{v}_{t-l}^u}(\bar{\mathbf{x}}_t^u)} + \frac{\xi_1' \sum_{h=1}^H [w_h \bar{f}_{v_m}^h(\bar{\mathbf{x}}_t^u)]}{\tau'_{\tilde{v}_{t-l}^u \tilde{v}_{t-l}^u}(\bar{\mathbf{x}}_t^u)} \right]$$

$$\leq \mathbb{P}\left[ \eta'_{p_2'}(\bar{\mathbf{x}}_t^u) \leq \eta'_{\tilde{v}_{t-l}^u}(\bar{\mathbf{x}}_t^u) < \eta'_{p_2'}(\bar{\mathbf{x}}_t^u) + \frac{\epsilon' + \xi_1'}{\tau'_{\tilde{v}_{t-l}^u \tilde{v}_{t-l}^u}(\bar{\mathbf{x}}_t^u)} \right].$$

$$(16)$$

Recall that $\epsilon' \leq \alpha_0 \tau'_{\tilde{v}_{t-l}^u \tilde{v}_{t-l}^u}(\bar{\mathbf{x}}_t^u) - \xi_1'$, which implies $\frac{\epsilon' + \xi_1'}{\tau'_{\tilde{v}_{t-l}^u \tilde{v}_{t-l}^u}(\bar{\mathbf{x}}_t^u)} \leq \alpha_0$. Hence, the relaxed Multiclass Tsybakov Condition holds, and the probability of the first term of $\boldsymbol{E}_{\mathrm{DDUI}}$ (Eq. 14) is bounded by $C\left( \frac{\epsilon' + \xi_1'}{\tau'_{\tilde{v}_{t-l}^u \tilde{v}_{t-l}^u}(\bar{\mathbf{x}}_t^u)} \right)^\lambda$, namely, $C\big(\mathcal{O}(\epsilon' + \xi_1')\big)^\lambda$.

For the second term of $\boldsymbol{E}_{\mathrm{DDUI}}$, we have:

$$\mathbb{P}\left[ p_1' \neq \tilde{v}_{t-l}^u, \frac{\sum_{h=1}^H [w_h \bar{f}_{\tilde{v}_{t-l}^u}^h(\bar{\mathbf{x}}_t^u)]}{\sum_{h=1}^H [w_h \bar{f}_{\tilde{v}_m}^h(\bar{\mathbf{x}}_t^u)]} \geq \beta' \right]$$

$$= \mathbb{P}\left[ p_1' \neq \tilde{v}_{t-l}^u, p_1' = \bar{v}_m, \frac{\sum_{h=1}^H [w_h \bar{f}_{\tilde{v}_{t-l}^u}^h(\bar{\mathbf{x}}_t^u)]}{\sum_{h=1}^H [w_h \bar{f}_{\tilde{v}_m}^h(\bar{\mathbf{x}}_t^u)]} \geq \beta' \right] \qquad (17)$$

$$+ \mathbb{P}\left[ p_1' \neq \tilde{v}_{t-l}^u, p_1' \neq \bar{v}_m, \frac{\sum_{h=1}^H [w_h \bar{f}_{\tilde{v}_{t-l}^u}^h(\bar{\mathbf{x}}_t^u)]}{\sum_{h=1}^H [w_h \bar{f}_{\tilde{v}_m}^h(\bar{\mathbf{x}}_t^u)]} \geq \beta' \right]$$

For the first term of Eq. 17, we have:

$$
\mathbb{P}\left[p_1' \neq \tilde{v}_{t-l}^u, p_1' = \breve{v}_m, \frac{\sum_{h=1}^{H}\left[w_h \overleftarrow{f}_{\tilde{v}_{t-l}^u}^h(\breve{\mathbf{x}}_t^u)\right]}{\sum_{h=1}^{H}\left[w_h \overleftarrow{f}_{\breve{v}_m}^h(\breve{\mathbf{x}}_t^u)\right]} \geq \beta'\right]
$$

$$
\leq \mathbb{P}\left[p_1' = \breve{v}_m, \frac{\sum_{h=1}^{H}\left[w_h \overleftarrow{f}_{\tilde{v}_{t-l}^u}^h(\breve{\mathbf{x}}_t^u)\right]}{\sum_{h=1}^{H}\left[w_h \overleftarrow{f}_{\breve{v}_m}^h(\breve{\mathbf{x}}_t^u)\right]} \geq \beta'\right]
$$

$$
\leq \mathbb{P}\left[p_1' = \breve{v}_m, \eta'_{\breve{v}_m}(\breve{\mathbf{x}}_t^u) \geq \eta'_{p_2'}(\breve{\mathbf{x}}_t^u), \sum_{h=1}^{H} w_h\left[\tilde{\eta}'_{\breve{v}_m}(\breve{\mathbf{x}}_t^u) - \epsilon'\right] \leq \frac{\sum_{h=1}^{H}\left[w_h \overleftarrow{f}_{\breve{v}_m}^h(\breve{\mathbf{x}}_t^u)\right]}{\beta'}\right]
$$

$$
= \mathbb{P}\left[p_1' = \breve{v}_m, \eta'_{\breve{v}_m}(\breve{\mathbf{x}}_t^u) \geq \eta'_{p_2'}(\breve{\mathbf{x}}_t^u), \tilde{\eta}'_{\breve{v}_m}(\breve{\mathbf{x}}_t^u) \leq \frac{\sum_{h=1}^{H}\left[w_h \overleftarrow{f}_{\breve{v}_m}^h(\breve{\mathbf{x}}_t^u)\right]}{\beta'} + \epsilon'\right]
$$

$$
= \mathbb{P}\left[p_1' = \breve{v}_m, \eta'_{\breve{v}_m}(\breve{\mathbf{x}}_t^u) \geq \eta'_{p_2'}(\breve{\mathbf{x}}_t^u), \sum_{v_j \in \mathcal{V}} \tau'_{v_j v_m}(\breve{\mathbf{x}}_t^u)\eta'_{v_j}(\breve{\mathbf{x}}_t^u) \leq \frac{\sum_{h=1}^{H}\left[w_h \overleftarrow{f}_{\breve{v}_m}^h(\breve{\mathbf{x}}_t^u)\right]}{\beta'} + \epsilon'\right]
$$

$$
\leq \mathbb{P}\left[p_1' = \breve{v}_m, \eta'_{p_2'}(\breve{\mathbf{x}}_t^u) \leq \eta'_{\breve{v}_m}(\breve{\mathbf{x}}_t^u) \leq \frac{\frac{\sum_{h=1}^{H}[w_h \overleftarrow{f}_{\breve{v}_m}^h(\breve{\mathbf{x}}_t^u)]}{\beta'} - \sum_{v_j \neq v_m}\left[\tau'_{v_j v_m}(\breve{\mathbf{x}}_t^u)\eta'_{v_j}(\breve{\mathbf{x}}_t^u)\right]}{\tau'_{\breve{v}_m \breve{v}_m}(\breve{\mathbf{x}}_t^u)} + \frac{\epsilon'}{\tau'_{\breve{v}_m \breve{v}_m}(\breve{\mathbf{x}}_t^u)}\right]
$$

$$
\leq \mathbb{P}\left[p_1' = \breve{v}_m, \eta'_{p_2'}(\breve{\mathbf{x}}_t^u) \leq \eta'_{\breve{v}_m}(\breve{\mathbf{x}}_t^u) \leq \frac{\frac{\sum_{h=1}^{H}[w_h \overleftarrow{f}_{\breve{v}_m}^h(\breve{\mathbf{x}}_t^u)]}{\beta_2' - \xi_2'} - \sum_{v_j \neq v_m}\left[\tau'_{v_j v_m}(\breve{\mathbf{x}}_t^u)\eta'_{v_j}(\breve{\mathbf{x}}_t^u)\right]}{\tau'_{\breve{v}_m \breve{v}_m}(\breve{\mathbf{x}}_t^u)} + \frac{\epsilon'}{\tau'_{\breve{v}_m \breve{v}_m}(\breve{\mathbf{x}}_t^u)}\right]
$$

$$
= \mathbb{P}\left[p_1' = \breve{v}_m, \eta'_{p_2'}(\breve{\mathbf{x}}_t^u) \leq \eta'_{\breve{v}_m}(\breve{\mathbf{x}}_t^u) \leq \frac{\frac{\sum_{h=1}^{H}[w_h \overleftarrow{f}_{\breve{v}_m}^h(\breve{\mathbf{x}}_t^u)]}{\beta_2'} - \sum_{v_j \neq v_m}\left[\tau'_{v_j v_m}(\breve{\mathbf{x}}_t^u)\eta'_{v_j}(\breve{\mathbf{x}}_t^u)\right]}{\tau'_{\breve{v}_m \breve{v}_m}(\breve{\mathbf{x}}_t^u)} + \frac{\epsilon'}{\tau'_{\breve{v}_m \breve{v}_m}(\breve{\mathbf{x}}_t^u)}\right.
$$
$$
\left. + \frac{\xi_2' \sum_{h=1}^{H}[w_h \overleftarrow{f}_{\breve{v}_m}^h(\breve{\mathbf{x}}_t^u)]}{\beta_2'(\beta_2' - \xi_2')\tau'_{\breve{v}_m \breve{v}_m}(\breve{\mathbf{x}}_t^u)}\right]
$$

(18)

We replace $\beta_2'$ with $\frac{\sum_{h=1}^{H}[w_h \overleftarrow{f}_{\tilde{v}_t^u}^h(\breve{\mathbf{x}}_t^u)]}{\tau'_{\breve{v}_m \breve{v}_m}(\breve{\mathbf{x}}_t^u)\eta'_{p_2'}(\breve{\mathbf{x}}_t^u)+\sum_{v_j \neq v_m}\tau'_{v_j v_m}(\breve{\mathbf{x}}_t^u)\eta'_{v_j}(\breve{\mathbf{x}}_t^u)}$ and continue the calculation:

$$
\mathbb{P}\left[p_1' \neq \tilde{v}_t^u, p_1' = v_m, \frac{\sum_{h=1}^{H}\left[w_h \overleftarrow{f}_{\tilde{v}_t^u}^h(\breve{\mathbf{x}}_t^u)\right]}{\sum_{h=1}^{H}\left[w_h \overleftarrow{f}_{\breve{v}_m}^h(\breve{\mathbf{x}}_t^u)\right]} \geq \beta\right]
$$

$$
\leq \mathbb{P}\left[p_1' = \breve{v}_m, \eta'_{p_2'}(\breve{\mathbf{x}}_t^u) \leq \eta'_{\breve{v}_m}(\breve{\mathbf{x}}_t^u) \leq \eta'_{p_2'}(\breve{\mathbf{x}}_t^u) + \frac{\epsilon'}{\tau'_{\breve{v}_m \breve{v}_m}(\breve{\mathbf{x}}_t^u)} + \frac{\xi_2' \sum_{h=1}^{H}[w_h \overleftarrow{f}_{\breve{v}_m}^h(\breve{\mathbf{x}}_t^u)]}{\beta_2'(\beta_2' - \xi_2')\tau'_{\breve{v}_m \breve{v}_m}(\breve{\mathbf{x}}_t^u)}\right] \quad (19)
$$

$$
\leq \mathbb{P}\left[\eta'_{p_2'}(\breve{\mathbf{x}}_t^u) \leq \eta'_{\breve{v}_m}(\breve{\mathbf{x}}_t^u) \leq \eta'_{p_2'}(\breve{\mathbf{x}}_t^u) + \frac{\epsilon'}{\tau'_{\breve{v}_m \breve{v}_m}(\breve{\mathbf{x}}_t^u)} + \frac{\xi_2'}{\beta_2'(\beta_2' - \xi_2')\tau'_{\breve{v}_m \breve{v}_m}(\breve{\mathbf{x}}_t^u)}\right].
$$

Recall that $\epsilon' \leq \frac{\alpha_0 \tau'_{\breve{v}_m \breve{v}_m}(\breve{\mathbf{x}}_t^u)\beta_2'(\beta_2' - \xi_2') - \xi_2'}{\beta_2'(\beta_2' - \xi_2')}$, which implies $\frac{\epsilon'}{\tau'_{\breve{v}_m \breve{v}_m}(\breve{\mathbf{x}}_t^u)} + \frac{\xi_2'}{\beta_2'(\beta_2' - \xi_2')\tau'_{\breve{v}_m \breve{v}_m}(\breve{\mathbf{x}}_t^u)} \leq \alpha_0$.
Hence, the relaxed Multiclass Tsybakov Condition holds and the probability of the first term of Eq. 17 is bounded by $C\left(\frac{\epsilon'\beta_2'(\beta_2' - \xi_2') + \xi_2'}{\beta_2'(\beta_2' - \xi_2')\tau'_{\breve{v}_m \breve{v}_m}(\breve{\mathbf{x}}_t^u)}\right)^\lambda$, namely, $C\left(\mathcal{O}(\epsilon' + \xi_2')\right)^\lambda$.

Finally, for the second term of Eq. 17, we have:

$$
\mathbb{P}\left[p_1' \neq \tilde{v}_{t-l}^u, p_1' \neq \bar{v}_m, \frac{\sum_{h=1}^H \left[w_h \overleftarrow{f}_{\tilde{v}_{t-l}^u}^h(\bar{\mathbf{x}}_t^u)\right]}{\sum_{h=1}^H \left[w_h \overleftarrow{f}_{\bar{v}_m}^h(\bar{\mathbf{x}}_t^u)\right]} \geq \beta'\right]
$$

$$
=\mathbb{P}\left[p_1' \neq v_m, \eta_{p_1'}'(\bar{\mathbf{x}}_t^u) \geq \eta_{p_2'}'(\bar{\mathbf{x}}_t^u), \frac{\sum_{h=1}^H \left[w_h \overleftarrow{f}_{p_1'}^h(\bar{\mathbf{x}}_t^u)\right]}{\sum_{h=1}^H \left[w_h \overleftarrow{f}_{\bar{v}_m}^h(\bar{\mathbf{x}}_t^u)\right]} < 1\right]
$$

$$
\leq\mathbb{P}\left[\eta_{p_1'}'(\bar{\mathbf{x}}_t^u) \geq \eta_{p_2'}'(\bar{\mathbf{x}}_t^u), \sum_{h=1}^H \left[w_h \overleftarrow{f}_{p_1'}^h(\bar{\mathbf{x}}_t^u)\right] < \sum_{h=1}^H \left[w_h \overleftarrow{f}_{\bar{v}_m}^h(\bar{\mathbf{x}}_t^u)\right]\right]
$$

$$
\leq\mathbb{P}\left[\eta_{p_1'}'(\bar{\mathbf{x}}_t^u) \geq \eta_{p_2'}'(\bar{\mathbf{x}}_t^u), \sum_{h=1}^H w_h \left[\tilde{\eta}_{p_1'}'(\bar{\mathbf{x}}_t^u) - \epsilon'\right] < \sum_{h=1}^H w_h \left[\tilde{\eta}_{\bar{v}_m}'(\bar{\mathbf{x}}_t^u) + \epsilon'\right]\right]
$$

$$
=\mathbb{P}\left[\eta_{p_1'}'(\bar{\mathbf{x}}_t^u) \geq \eta_{p_2'}'(\bar{\mathbf{x}}_t^u), \tilde{\eta}_{p_1'}'(\bar{\mathbf{x}}_t^u) - \epsilon' < \tilde{\eta}_{\bar{v}_m}'(\bar{\mathbf{x}}_t^u) + \epsilon'\right]
$$

$$
=\mathbb{P}\left[\eta_{p_1'}'(\bar{\mathbf{x}}_t^u) \geq \eta_{p_2'}'(\bar{\mathbf{x}}_t^u), \sum_{v_j \in \mathcal{V}} \tau_{v_j p_1'}'(\bar{\mathbf{x}}_t^u)\eta_{v_j}'(\bar{\mathbf{x}}_t^u) < \sum_{v_j \in \mathcal{V}} \tau_{\bar{v}_m}'(\bar{\mathbf{x}}_t^u)\eta_{v_j}'(\bar{\mathbf{x}}_t^u) + 2\epsilon'\right]
$$

(20)

$$
=\mathbb{P}\left[\eta_{p_1'}'(\bar{\mathbf{x}}_t^u) \geq \eta_{p_2'}'(\bar{\mathbf{x}}_t^u), \tau_{p_1' p_1'}'(\bar{\mathbf{x}}_t^u)\eta_{p_1'}'(\bar{\mathbf{x}}_t^u) + \sum_{v_j \neq p_1'} \tau_{v_j p_1'}'(\bar{\mathbf{x}}_t^u)\eta_{v_j}'(\bar{\mathbf{x}}_t^u) < \right.
$$

$$
\left. \tau_{p_1' \bar{v}_m}'(\bar{\mathbf{x}}_t^u)\eta_{p_1'}'(\bar{\mathbf{x}}_t^u) + \sum_{v_j \neq p_1'} \tau_{\bar{v}_m}'(\bar{\mathbf{x}}_t^u)\eta_{v_j}'(\bar{\mathbf{x}}_t^u) + 2\epsilon'\right]
$$

$$
=\mathbb{P}\left[\eta_{p_1'}'(\bar{\mathbf{x}}_t^u) \geq \eta_{p_2'}'(\bar{\mathbf{x}}_t^u), \eta_{p_1'}'(\bar{\mathbf{x}}_t^u)\left[\tau_{p_1' p_1'}'(\bar{\mathbf{x}}_t^u) - \tau_{p_1' \bar{v}_m}'(\bar{\mathbf{x}}_t^u)\right] < \right.
$$

$$
\left. \sum_{v_j \neq p_1'} \left[\tau_{\bar{v}_m}'(\bar{\mathbf{x}}_t^u) - \tau_{v_j p_1'}'(\bar{\mathbf{x}}_t^u)\right]\eta_{v_j}'(\bar{\mathbf{x}}_t^u) + 2\epsilon'\right]
$$

$$
=\mathbb{P}\left[\eta_{p_2'}'(\bar{\mathbf{x}}_t^u) \leq \eta_{p_1'}'(\bar{\mathbf{x}}_t^u) < \eta_{p_2'}'(\bar{\mathbf{x}}_t^u) + \right.
$$

$$
\left. \frac{\sum_{v_j \neq p_1'} \left[\tau_{\bar{v}_m}'(\bar{\mathbf{x}}_t^u) - \tau_{v_j p_1'}'(\bar{\mathbf{x}}_t^u)\right]\eta_{v_j}'(\bar{\mathbf{x}}_t^u) + 2\epsilon' - \left[\tau_{p_1' p_1'}'(\bar{\mathbf{x}}_t^u) - \tau_{p_1' \bar{v}_m}'(\bar{\mathbf{x}}_t^u)\right]\eta_{p_2'}'(\bar{\mathbf{x}}_t^u)}{\left[\tau_{p_1' p_1'}'(\bar{\mathbf{x}}_t^u) - \tau_{p_1' \bar{v}_m}'(\bar{\mathbf{x}}_t^u)\right]}\right]
$$

Recall that $\epsilon' \leq \frac{1}{2}\left[\left[\tau_{p_1' p_1'}'(\bar{\mathbf{x}}_t^u) - \tau_{p_1' \bar{v}_m}'(\bar{\mathbf{x}}_t^u)\right][\alpha_0 + \eta_{p_2'}'(\bar{\mathbf{x}}_t^u)] - \sum_{v_j \neq p_1'}[\tau_{\bar{v}_m}'(\bar{\mathbf{x}}_t^u) - \tau_{v_j p_1'}'(\bar{\mathbf{x}}_t^u)]\eta_{v_j}'(\bar{\mathbf{x}}_t^u)\right]$, which implies $\frac{\sum_{v_j \neq p_1'} \left[\tau_{\bar{v}_m}'(\bar{\mathbf{x}}_t^u) - \tau_{v_j p_1'}'(\bar{\mathbf{x}}_t^u)\right]\eta_{v_j}'(\bar{\mathbf{x}}_t^u) + 2\epsilon' - \left[\tau_{p_1' p_1'}'(\bar{\mathbf{x}}_t^u) - \tau_{p_1' \bar{v}_m}'(\bar{\mathbf{x}}_t^u)\right]\eta_{p_2'}'(\bar{\mathbf{x}}_t^u)}{\left[\tau_{p_1' p_1'}'(\bar{\mathbf{x}}_t^u) - \tau_{p_1' \bar{v}_m}'(\bar{\mathbf{x}}_t^u)\right]} \leq \alpha_0$. Hence, the relaxed Multiclass Tsybakov Condition holds and the probability of Case-3 is bounded by $C\left(\frac{\sum_{v_j \neq p_1'} \left[\tau_{\bar{v}_m}'(\bar{\mathbf{x}}_t^u) - \tau_{v_j p_1'}'(\bar{\mathbf{x}}_t^u)\right]\eta_{v_j}'(\bar{\mathbf{x}}_t^u) + 2\epsilon' - \left[\tau_{p_1' p_1'}'(\bar{\mathbf{x}}_t^u) - \tau_{p_1' \bar{v}_m}'(\bar{\mathbf{x}}_t^u)\right]\eta_{p_2'}'(\bar{\mathbf{x}}_t^u)}{\left[\tau_{p_1' p_1'}'(\bar{\mathbf{x}}_t^u) - \tau_{p_1' \bar{v}_m}'(\bar{\mathbf{x}}_t^u)\right]}\right)^\lambda$, namely, $C(\mathcal{O}(\epsilon'))^\lambda$. $\quad\square$

### B.5 The Proof of Theorem 4

**Theorem 4.** *Let $\mathcal{R}$ be a list of $K$ items randomly sampled from $\mathcal{V}$ with replacement, $\zeta \in (0,1)$, $r_H(\mathbf{x}_t^u, v_i) = \sum_{v_j \in \mathcal{V}} \mathbb{I}\left[\sum_{h=1}^H [w_h f_{v_j}^h(\bar{\mathbf{x}}_t^u)] > \sum_{h=1}^H [w_h f_{v_i}^h(\bar{\mathbf{x}}_t^u)]\right]$ be the rank of item $v_i$ over the entire item set at the $H$-th epoch, and $\hat{r}_H(\mathbf{x}_t^u, v_i) = \sum_{v_j \in \mathcal{R}} \mathbb{I}\left[\sum_{h=1}^H [w_h f_{v_j}^h(\bar{\mathbf{x}}_t^u)] > \sum_{h=1}^H [w_h f_{v_i}^h(\bar{\mathbf{x}}_t^u)]\right]$ be the rank of $v_i$ over $\mathcal{R}$ at the $H$-th epoch. We have: $\mathbb{P}\left[\left|\frac{\hat{r}_H(\mathbf{x}_t^u, v_i)}{K} - \frac{r_H(\mathbf{x}_t^u, v_i)}{|\mathcal{V}|}\right| \geq \zeta\right] \leq \exp(-2K\zeta^2)$.*

*Proof.* $\hat{r}_H(\mathbf{x}_t^u, v_i)$ can be deemed as a random variable that counts the number of items ranked higher than $v_i$ in $\mathcal{R}$. Since the items in $\mathcal{R}$ are randomly sampled from $\mathcal{V}$ with replacement, $\hat{r}_H(\mathbf{x}_t^u, v_i)$ follows a Binomial distribution $\mathrm{Binomial}(K, \frac{r_H(\mathbf{x}_t^u, v_i)}{|\mathcal{V}|})$. Thus, we have:

$$
\mathbb{E}\left[\hat{r}_H(\mathbf{x}_t^u, v_i)\right] = \frac{K \cdot r_H(\mathbf{x}_t^u, v_i)}{|\mathcal{V}|}.
$$

(21)

Meanwhile, $\hat{r}_H(\mathbf{x}_t^u, v_i)$ can also be viewed as the sum of $K$ i.i.d. Bernoulli random variables $b_1, b_2, ..., b_k$, where $b_k = 1(1 \leq k \leq K)$ indicates the $k$-th sampled item for $\mathcal{R}$ is ranked higher than $v_i$, and $b_k = 0$ otherwise. Hence, by applying Hoeffding's inequality [5], we have:

$$\mathbb{P}\left[\left|\frac{1}{K}\sum_{k=1}^{K} b_k - \mathbb{E}\left[\frac{1}{K}\sum_{k=1}^{K} b_k\right]\right| \geq \zeta\right] \leq \exp(-2K\zeta^2). \tag{22}$$

Then by replacing $\sum_{k=1}^{K} b_k$ with $\hat{r}_H(\mathbf{x}_t^u, v_i)$, we obtain:

$$\mathbb{P}\left[\left|\frac{\hat{r}_H(\mathbf{x}_t^u, v_i)}{K} - \frac{\mathbb{E}\left[\hat{r}_H(\mathbf{x}_t^u, v_i)\right]}{K}\right| \geq \zeta\right] \leq \exp(-2K\zeta^2). \tag{23}$$

Next, we replace $\mathbb{E}\left[\hat{r}_H(\mathbf{x}_t^u, v_i)\right]$ with $\frac{K \cdot r_H(\mathbf{x}_t^u, v_i)}{|\mathcal{V}|}$ according to Eq. 21:

$$\mathbb{P}\left[\left|\frac{\hat{r}_H(\mathbf{x}_t^u, v_i)}{K} - \frac{r_H(\mathbf{x}_t^u, v_i)}{|\mathcal{V}|}\right| \geq \zeta\right] \leq \exp(-2K\zeta^2). \tag{24}$$

$\square$

## C   Additional Experimental Results

### C.1   Hyper-parameter Settings for Baselines

For fair comparisons, we implement FPMC with PyTorch. For other baselines, we use the original code provided by the corresponding authors. All the baselines adopt Xavier [6] initializer and Adam [7] optimizer. We empirically find the optimal hyper-parameter setting for each baseline based on the performance on the validation set. The detailed hyper-parameter setting for each baseline is summarized in Table 1.

### C.2   More Results on Hyper-parameter Analysis

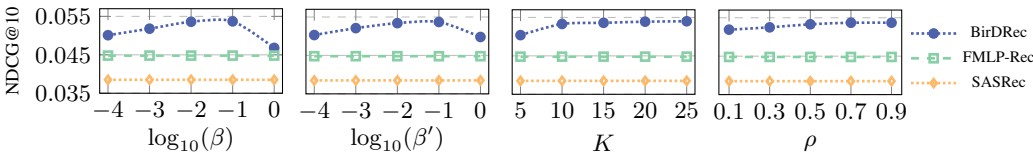

Figure 2: Effect of key hyper-parameters $\beta$, $\beta'$, $K$, and $\rho$ of BirDRec on Beauty dataset.

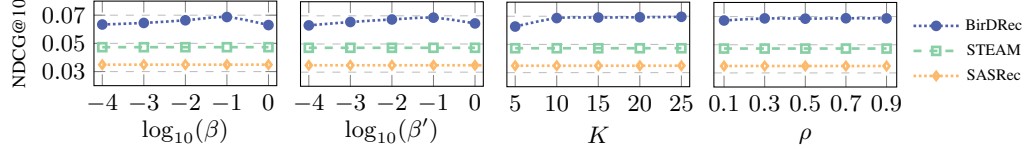

Figure 3: Effect of key hyper-parameters $\beta$, $\beta'$, $K$, and $\rho$ of BirDRec on Yelp dataset.

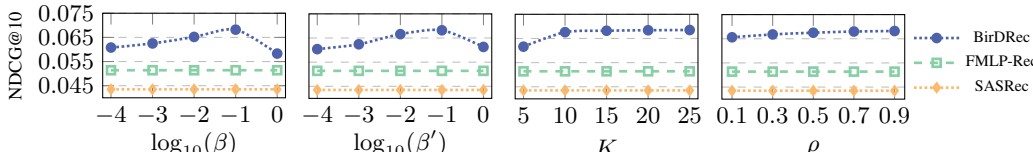

Figure 4: Effect of key hyper-parameters $\beta$, $\beta'$, $K$, and $\rho$ of BirDRec on QK-Vedio dataset.

### C.3   More Results on the Percentage of Rectified Instances

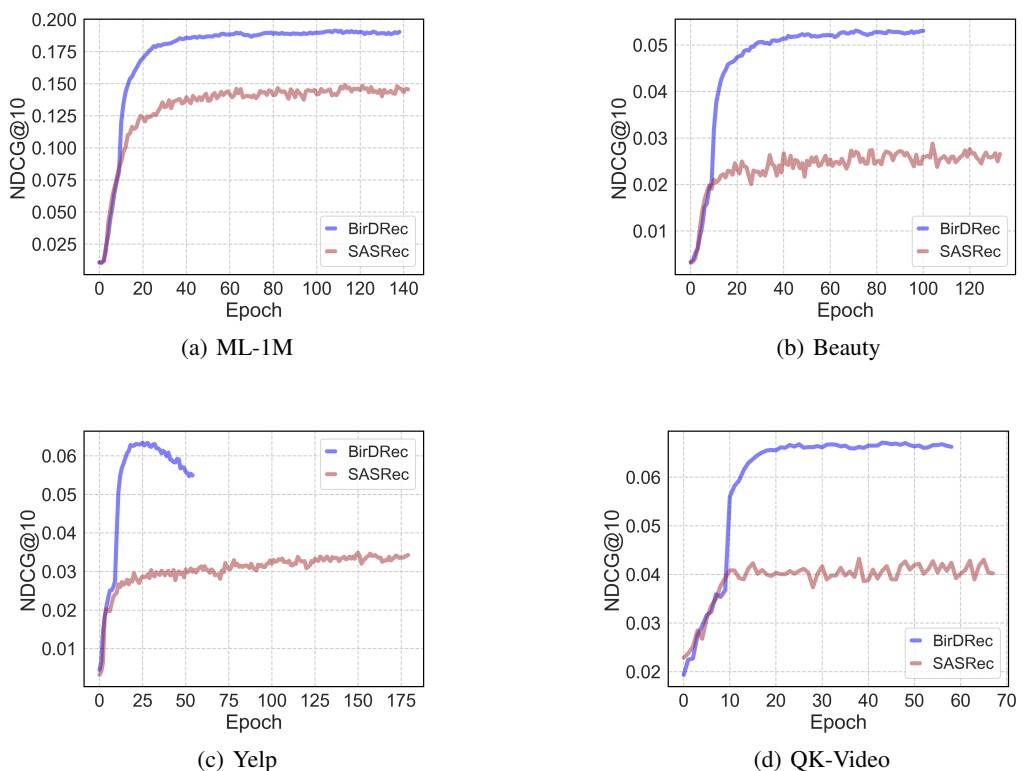

(a) ML-1M

(b) Beauty

(c) Yelp

(d) QK-Video

Figure 5: Testing accuracy (NDCG@10) of our BirDRec and SASRec with increasing epochs. The training stops if the best accuracy does not increase in 25 consecutive epochs.

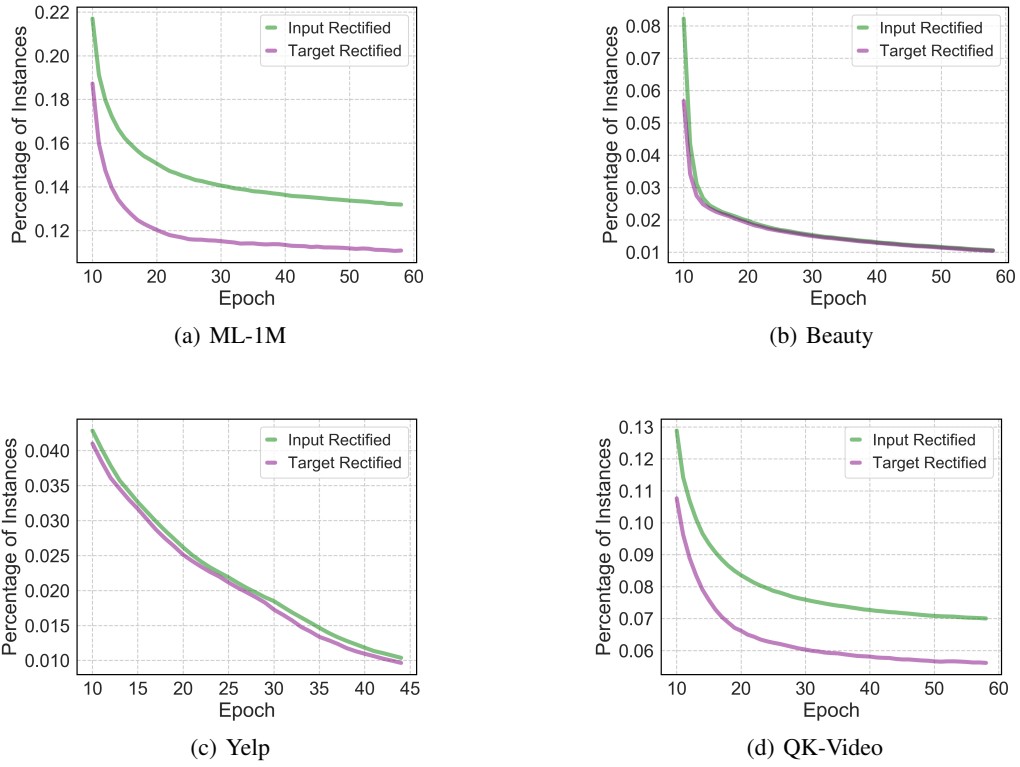

(a) ML-1M

(b) Beauty

(c) Yelp

(d) QK-Video

Figure 6: The percentage of instances that are rectified with increasing epochs.

Table 1: The search space of hyper-parameters and the optimal settings found by grid search for all baselines on the four real-world datasets.

| | Parameter | ML | Be | Ye | QK | Search Space |
|---|---|---|---|---|---|---|
| **FPMC** | embedding_size | 64 | 64 | 64 | 128 | $\{16, 32, 64, 128\}$ |
| | learning_rate | $10^{-3}$ | $10^{-3}$ | $10^{-2}$ | $10^{-3}$ | $\{10^{-4}, 10^{-3}, 10^{-2}, 10^{-1}\}$ |
| | L2_regularization_coefficient | $10^{-2}$ | $10^{-3}$ | $10^{-3}$ | $10^{-2}$ | $\{10^{-4}, 10^{-3}, 10^{-2}, 10^{-1}\}$ |
| | input_length | 10 | 5 | 5 | 10 | $\{5, 10, 20, 30, 40, 50\}$ |
| | batch_size | 1024 | 512 | 512 | 1024 | $\{128, 256, 512, 1024\}$ |
| **Caser** | embedding_size | 64 | 64 | 64 | 128 | $\{16, 32, 64, 128\}$ |
| | learning_rate | $10^{-3}$ | $10^{-3}$ | $10^{-2}$ | $10^{-3}$ | $\{10^{-4}, 10^{-3}, 10^{-2}, 10^{-1}\}$ |
| | L2_regularization_coefficient | $10^{-2}$ | $10^{-3}$ | $10^{-3}$ | $10^{-2}$ | $\{10^{-4}, 10^{-3}, 10^{-2}, 10^{-1}\}$ |
| | input_length | 5 | 5 | 5 | 5 | $\{5, 10, 20, 30, 40, 50\}$ |
| | batch_size | 512 | 256 | 256 | 512 | $\{128, 256, 512, 1024\}$ |
| | horizontal_filter_num | 16 | 16 | 16 | 32 | $\{4, 8, 16, 32, 64\}$ |
| | vertical_filter_num | 4 | 4 | 4 | 8 | $\{1, 2, 4, 8, 16\}$ |
| **GRU4Rec** | embedding_size | 64 | 64 | 64 | 128 | $\{16, 32, 64, 128\}$ |
| | learning_rate | $10^{-2}$ | $10^{-2}$ | $10^{-2}$ | $10^{-2}$ | $\{10^{-4}, 10^{-3}, 10^{-2}, 10^{-1}\}$ |
| | L2_regularization_coefficient | $10^{-2}$ | $10^{-2}$ | $10^{-2}$ | $10^{-2}$ | $\{10^{-4}, 10^{-3}, 10^{-2}, 10^{-1}\}$ |
| | input_length | 10 | 5 | 5 | 10 | $\{5, 10, 20, 30, 40, 50\}$ |
| | batch_size | 1024 | 512 | 512 | 1024 | $\{128, 256, 512, 1024\}$ |
| | GRU_unit_number | 256 | 256 | 256 | 512 | $\{128, 256, 512, 1024\}$ |
| **SASRec** | embedding_size | 64 | 64 | 64 | 128 | $\{16, 32, 64, 128\}$ |
| | learning_rate | $10^{-2}$ | $10^{-2}$ | $10^{-2}$ | $10^{-3}$ | $\{10^{-4}, 10^{-3}, 10^{-2}, 10^{-1}\}$ |
| | L2_regularization_coefficient | $10^{-2}$ | $10^{-2}$ | $10^{-2}$ | $10^{-2}$ | $\{10^{-4}, 10^{-3}, 10^{-2}, 10^{-1}\}$ |
| | input_length | 50 | 20 | 20 | 40 | $\{5, 10, 20, 30, 40, 50\}$ |
| | batch_size | 128 | 128 | 512 | 1024 | $\{128, 256, 512, 1024\}$ |
| | self_attention_head_num | 2 | 1 | 2 | 4 | $\{1, 2, 4, 8\}$ |
| | self_attention_block_num | 1 | 1 | 1 | 2 | $\{1, 2, 3, 4\}$ |
| **BERT4Rec** | embedding_size | 64 | 64 | 64 | 128 | $\{16, 32, 64, 128\}$ |
| | learning_rate | $10^{-4}$ | $10^{-4}$ | $10^{-2}$ | $10^{-3}$ | $\{10^{-4}, 10^{-3}, 10^{-2}, 10^{-1}\}$ |
| | L2_regularization_coefficient | $10^{-2}$ | $10^{-2}$ | $10^{-2}$ | $10^{-2}$ | $\{10^{-4}, 10^{-3}, 10^{-2}, 10^{-1}\}$ |
| | input_length | 50 | 20 | 20 | 40 | $\{5, 10, 20, 30, 40, 50\}$ |
| | batch_size | 256 | 256 | 512 | 1024 | $\{128, 256, 512, 1024\}$ |
| | self_attention_head_num | 4 | 1 | 2 | 4 | $\{1, 2, 3, 4\}$ |
| | self_attention_block_num | 4 | 1 | 1 | 2 | $\{1, 2, 3, 4\}$ |
| **MAGNN** | embedding_size | 64 | 64 | 64 | 128 | $\{16, 32, 64, 128\}$ |
| | learning_rate | $10^{-3}$ | $10^{-3}$ | $10^{-2}$ | $10^{-3}$ | $\{10^{-4}, 10^{-3}, 10^{-2}, 10^{-1}\}$ |
| | L2_regularization_coefficient | $10^{-2}$ | $10^{-2}$ | $10^{-2}$ | $10^{-2}$ | $\{10^{-4}, 10^{-3}, 10^{-2}, 10^{-1}\}$ |
| | input_length | 5 | 5 | 5 | 5 | $\{5, 10, 20, 30, 40, 50\}$ |
| | batch_size | 1024 | 512 | 512 | 1024 | $\{128, 256, 512, 1024\}$ |
| | GNN_layer_num | 2 | 2 | 2 | 2 | $\{1, 2, 3, 4\}$ |
| | memory_unit_num | 10 | 10 | 10 | 20 | $\{5, 10, 15, 20\}$ |
| **BERD** | embedding_size | 64 | 64 | 64 | 128 | $\{16, 32, 64, 128\}$ |
| | learning_rate | $10^{-3}$ | $10^{-3}$ | $10^{-2}$ | $10^{-3}$ | $\{10^{-4}, 10^{-3}, 10^{-2}, 10^{-1}\}$ |
| | L2_regularization_coefficient | $10^{-2}$ | $10^{-2}$ | $10^{-2}$ | $10^{-2}$ | $\{10^{-4}, 10^{-3}, 10^{-2}, 10^{-1}\}$ |
| | input_length | 5 | 5 | 5 | 5 | $\{5, 10, 20, 30, 40, 50\}$ |
| | batch_size | 1024 | 512 | 512 | 1024 | $\{128, 256, 512, 1024\}$ |
| | self_attention_head_num | 2 | 2 | 2 | 2 | $\{1, 2, 3, 4\}$ |
| | self_attention_block_num | 1 | 1 | 1 | 2 | $\{1, 2, 3, 4\}$ |
| | UGCN_layer_num | 2 | 2 | 2 | 2 | $\{1, 2, 3, 4\}$ |
| | filter_ratio | 0.05 | 0.05 | 0.05 | 0.10 | $\{0.05, 0.10, 0.15, 0.20, 0.25\}$ |
| | sample_size of $\mathcal{L}_{sam}$ | 4 | 4 | 4 | 4 | $\{1, 2, 3, 4\}$ |
| **FMLP-Rec** | embedding_size | 64 | 64 | 64 | 128 | $\{16, 32, 64, 128\}$ |
| | learning_rate | $10^{-3}$ | $10^{-3}$ | $10^{-3}$ | $10^{-3}$ | $\{10^{-4}, 10^{-3}, 10^{-2}, 10^{-1}\}$ |
| | L2_regularization_coefficient | $10^{-2}$ | $10^{-3}$ | $10^{-3}$ | $10^{-2}$ | $\{10^{-4}, 10^{-3}, 10^{-2}, 10^{-1}\}$ |
| | input_length | 50 | 20 | 20 | 40 | $\{5, 10, 20, 30, 40, 50\}$ |
| | batch_size | 1024 | 256 | 256 | 1024 | $\{128, 256, 512, 1024\}$ |
| | learnable_filter_block_num | 2 | 2 | 2 | 2 | $\{1, 2, 3, 4\}$ |
| **STEAM** | embedding_size | 64 | 64 | 64 | 128 | $\{16, 32, 64, 128\}$ |
| | learning_rate | $10^{-3}$ | $10^{-3}$ | $10^{-2}$ | $10^{-3}$ | $\{10^{-4}, 10^{-3}, 10^{-2}, 10^{-1}\}$ |
| | L2_regularization_coefficient | $10^{-2}$ | $10^{-3}$ | $10^{-3}$ | $10^{-2}$ | $\{10^{-4}, 10^{-3}, 10^{-2}, 10^{-1}\}$ |
| | input_length | 50 | 20 | 20 | 40 | $\{5, 10, 20, 30, 40, 50\}$ |
| | batch_size | 1024 | 256 | 256 | 1024 | $\{128, 256, 512, 1024\}$ |
| | self_attention_head_num | 2 | 1 | 1 | 2 | $\{1, 2, 3, 4\}$ |
| | self_attention_block_num | 1 | 1 | 1 | 2 | $\{1, 2, 3, 4\}$ |
| | insertion_probability | 0.2 | 0.4 | 0.4 | 0.2 | $\{0.1, 0.2, 0.3, 0.4, 0.5\}$ |
| | deletion_probability | 0.1 | 0.1 | 0.1 | 0.1 | $\{0.1, 0.2, 0.3, 0.4, 0.5\}$ |
| | mask_probability | 0.4 | 0.5 | 0.5 | 0.3 | $\{0.1, 0.2, 0.3, 0.4, 0.5\}$ |
| **BirDRec** | embedding_size | 64 | 64 | 64 | 64 | $\{16, 32, 64, 128\}$ |
| | learning_rate | $10^{-3}$ | $10^{-3}$ | $10^{-2}$ | $10^{-3}$ | $\{10^{-4}, 10^{-3}, 10^{-2}, 10^{-1}\}$ |
| | L2_regularization_coefficient | $10^{-2}$ | $10^{-2}$ | $10^{-2}$ | $10^{-2}$ | $\{10^{-4}, 10^{-3}, 10^{-2}, 10^{-1}\}$ |
| | input_length | 5 | 5 | 5 | 5 | $\{5, 10, 20, 30, 40, 50\}$ |
| | batch_size | 1024 | 1024 | 1024 | 1024 | $\{128, 256, 512, 1024\}$ |
| | threshold $\beta$ in DRUT | $10^{-1}$ | $10^{-1}$ | $10^{-1}$ | $10^{-1}$ | $\{10^{-4}, 10^{-3}, 10^{-2}, 10^{-1}, 10^{0}\}$ |
| | threshold $\beta'$ in DDUI | $10^{-1}$ | $10^{-1}$ | $10^{-1}$ | $10^{-1}$ | $\{10^{-4}, 10^{-3}, 10^{-2}, 10^{-1}, 10^{0}\}$ |
| | size $K$ of rectification pools | 10 | 10 | 10 | 10 | $\{5, 10, 15, 20, 25\}$ |
| | exponential decay rate $\rho$ | 0.9 | 0.9 | 0.9 | 0.9 | $\{0.1, 0.3, 0.5, 0.7, 0.9\}$ |

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
