# OpenReview forum: "Theoretically Guaranteed Bidirectional Data Rectification for Robust Sequential Recommendation"
_NeurIPS.cc/2023/Conference — NeurIPS 2023 poster_

### Official Review · Reviewer_Uqk2 · 2023-07-04

**Soundness:** 4 excellent
**Presentation:** 3 good
**Contribution:** 4 excellent
**Rating:** 8
**Confidence:** 4

**Summary:**

Overall comment:
This paper proposes two detect and rectify methods to solve the problem of unreliable instances in Sequential Recommender Systems (SRSs), including unreliable items in the input and unreliable targets. The two methods can be applied to existing SRSs models, effectively improving the model performance.

Strengths of the paper include:

1. The paper provides sufficient theoretical proof for the methods, which is very impressive.
2. The discussion and clarification of related work, as well as the differences and connections with existing work, are very clear.
3. The experimental results are sufficient, and the original model performance is improved on multiple baseline models via proposed method. The ablation study also compares the performance of the DRUT, DDUI, their combination, and whether to use self-ensemble under different conditions.

Weakness of the article includes:

The paper does not discuss the limitations of the methods. In particular, I am interested in whether this method can be applied in large-scale real-world recommender systems? Are there any practical issues?

**Strengths:**

Strengths of the paper include:

1. The paper provides sufficient theoretical proof for the methods, which is very impressive.
2. The discussion and clarification of related work, as well as the differences and connections with existing work, are very clear.
3. The experimental results are sufficient, and the original model performance is improved on multiple baseline models via proposed method. The ablation study also compares the performance of the DRUT, DDUI, their combination, and whether to use self-ensemble under different conditions.

**Weaknesses:**

Weakness of the article includes:

The paper does not discuss the limitations of the methods. In particular, I am interested in whether this method can be applied in large-scale real-world recommender systems? Are there any practical issues?

**Questions:**

 I am interested in whether this method can be applied in large-scale real-world recommender systems? Are there any practical issues?

**Limitations:**

As I written in weakness:

The paper does not discuss the limitations of the methods. In particular, I am interested in whether this method can be applied in large-scale real-world recommender systems? Are there any practical issues?

---

> ### Author Rebuttal · Authors · 2023-08-09
>
>
> We sincerely appreciate your appreciation on our work and your valuable reviews.
>
> > Question and Limitation: the paper does not discuss the limitations of the methods. In particular, I am interested in whether this method can be applied in large-scale real-world recommender systems. Are there any practical issues?
>
> Thank you for highlighting the need to discuss the limitations and practical considerations of our proposed BirDRec framework. The main limitations are twofold, which will be highlighted in our camera-ready.
>
>
> - **Limited improvements on storage cost for smaller item sets:** As mentioned in lines 293-295, we acknowledge that the storage cost of BirDRec is marginally higher than STEAM's on ML-1M. This is primarily because the rectification sampling strategy with Self-ensemble has a less apparent advantage on datasets with smaller item sets (see lines 246-249). To be specific, the storage cost reduction for calculating weighted average prediction scores is from $O(|V|*H)$ to $O(K)$ for each instance. Thus if $|V|$ is small, the benefit of this reduction will be less obvious.
> - **Marginally higher computational cost than the backbone:** Although BirDRec is significantly faster than the latest robust SRS (STEAM), it is worth noting that BirDRec is 1.6 times on average slower than its backbone model SASRec (as depicted in Fig 3) in each training epoch. This increased training time could be a practical concern in systems with **extremely large-scale** datasets and real-time recommendation demands.

---

> > ### Comment · Reviewer_Uqk2 · 2023-08-20
> > **Reply to author rebuttal**
> >
> > I have no further questions. The limitations of the work are discussed.

---

### Official Review · Reviewer_cUYC · 2023-07-05

**Soundness:** 3 good
**Presentation:** 3 good
**Contribution:** 3 good
**Rating:** 7
**Confidence:** 2

**Summary:**

This paper aims to address the issue where the performance of sequential recommender systems (SRSs) could be harmed by mismatched input-target pairs introduced by distracted users. The paper claims that existing studies only focus on either unreliable inputs or targets but not both simultaneously. Also no theoretical guarantee of effectiveness for the existing methods are provided.  To address these issues, the paper proposes to replace/remove target/input item using a forward/backward SRS's weighted average scores accumulated over training epochs. Error bounds for both forward and backward rectifications are provided. To further reduce the time and space complexity, sampling strategy as well as self-ensembled mechanism are applied. Experiments show the proposed approach is able to improve the performance is representative SBSs on four datasets. Comparing to rectified/denoised SBSs, the performance of the proposed approach is also reported to be better. Ablation test as well as hyperparameter analysis are also provided.

**Strengths:**

* The ideas of a. rectifying input and target items using weighted average prediction score of forward and backward SBR, b. using sampling and self-ensembled mechanism of improving efficiency are interesting and new in my opinion.
* The main steps leading to the error bounds are reasonable, though I did not manage to check the proofs in the supplementary in details.
* The empirical studies that support the effectiveness of the propose approach appeals to be convincing.

**Weaknesses:**

As I am not familiar with the literature of the robustness of SRS nor the rectification / denoising in SRS, I am only able to give a few suggestions:
* Analyzing the existing work as well as the proposed approach on a synthetic dataset with ground truth could further strengthen the claims about: a. the drawbacks of the existing studies, b. the effectiveness of the proposed approach.  Conduct simulation on the synthetic dataset could validate the error bound, as well as evaluate the approximation quality of the proposed method.
* Qualitatively evaluating the rectified items for a few instances in different dataset would further help the audience understanding the behavior of the proposed approach.

**Questions:**

Please see the suggestions in the "weakness" section.

**Limitations:**

No limitation is mentioned in the paper.

---

> ### Author Rebuttal · Authors · 2023-08-09
>
>
> We appreciate your recognition of our work and your valuable feedback.
>
>
> > Weakness 1
>
>
> Thank you for the interesting and insightful suggestions.
> Constructing a synthetic dataset with ground truth for data reliability is indeed of importance for robust SRSs. However, the major challenge is how to obtain ground truth annotations. There are **NO definitive ground truth rules** that can be used to accurately identify the reliability of each instance due to the inherent ambiguity of user interactions. That is, the reason behind each interaction may not be apparent even to the users themselves, as noted in existing research [1]. Thus the annotation can not be done directly via either machine or crowdsourcing or domain experts.
>
> To this end, one possible solution is to design our own rules that are as close to the ground truth rules as possible for determining data reliability. However, we have to overcome key concerns including but not limited to the following ones:
>
> - **C1: How to guarantee the validity of the designed rules?** For example, we may design a rule: for users who are huge fans of Sci-Fi movies, if any instance of these users involves films of other genres, then the instance is unreliable. However, we may not be able to guarantee the generalizability of this rule to all these users without exceptions in the real world. So, it is challenging to quantitatively measure to what extent each rule is valid, leading to concerns about the validity of the synthetic data.
> - **C2: How to guarantee the completeness of the designed rules?** Considering the diverse user behavior patterns, it is impractical to design rules to exactly cover all the possible behavior patterns of all users in a specific domain. In this case, the synthetic data may either oversimplify or complicate the recommendation task, rendering the performance of SRSs on such datasets less meaningful.
>
> In conclusion, we acknowledge the invaluableness of the suggestions as well as the challenges involved. We will strive to make progress in this direction as part of our future research.
>
>
> > Weakness 2
>
>
> Thank you for the invaluable comments, which allow an intuitive and detailed analysis of the proposed method. We will randomly sample some unreliable instances on different datasets and visualize the rectification process and results. These will be added in the camera-ready.
>
> Reference
>
> [1] Cosley et al. Is seeing believing? How recommender system interfaces affect users’ opinions. In SIGCHI 2003.

---

> > ### Comment · Reviewer_cUYC · 2023-08-19
> >
> > Thanks for the explanations, I have no further questions.

---

### Official Review · Reviewer_1jEu · 2023-07-07

**Soundness:** 3 good
**Presentation:** 3 good
**Contribution:** 2 fair
**Rating:** 5
**Confidence:** 3

**Summary:**

This paper proposes a bidirectional data rectification framework, called BirDRec, to address the challenges of training sequential recommender systems (SRSs) with unreliable input and targets. The authors provide two theoretically guaranteed error-bounded strategies to rectify the data, which can be flexibly implemented with most existing SRSs. They also introduce a rectification sampling strategy and self-ensemble mechanism to reduce the computational and space complexity of BirDRec. The proposed framework is evaluated on several benchmark datasets, and the results demonstrate its generality, effectiveness, and efficiency compared to state-of-the-art robust SRSs.

**Strengths:**

1. The paper introduces BirDRec, a novel bidirectional data rectification framework designed to enhance the robustness of sequential recommender systems (SRSs) by addressing unreliable data. BirDRec can seamlessly integrate with existing SRSs, offering a flexible solution for improving their performance.
2. The authors present two error-bounded strategies within BirDRec that provide theoretical guarantees for detecting and rectifying unreliable input and targets in SRSs. These strategies contribute to the reliability and accuracy of the recommendations generated by the system.


**Weaknesses:**

1. The evaluation of the proposed framework is limited to a few benchmark datasets, and it remains uncertain how well it would generalize to different types of data or domains. It would be beneficial for the authors to include further experiments or discuss potential challenges in applying their framework to diverse datasets.
2. While the authors attribute the unreliable data to external distractions, it is important to acknowledge that there might be other sources of noise or errors that their approach does not account for. Including a discussion on potential alternative causes of unreliable data would strengthen the paper's analysis.
3. The paper lacks a thorough analysis of the computational and space complexity of the proposed framework. Understanding the resource requirements of the framework is crucial for assessing its scalability to larger datasets or real-time applications. It would be valuable for the authors to provide insights into the framework's efficiency and discuss any potential limitations in terms of computational resources.


**Questions:**

1. Could you provide additional details on the theoretical guarantees of the error-bounded strategies proposed for data rectification?
2. Can you explain the process behind selecting the benchmark datasets used in your experiments? What criteria did you consider when choosing these datasets? Additionally, could you elaborate on the evaluation metrics or performance measures used to assess the effectiveness of the proposed framework on these benchmarks?
3. Have you taken into account other sources of noise or errors commonly encountered in SRSs, such as cold-start, diversity, or fairness issues? If so, how does your approach address or mitigate these challenges?
4. In real-world applications, how do you envision the proposed framework being utilized? Are there specific domains or use cases where you believe BirDRec would be particularly beneficial? Additionally, what are some potential challenges or limitations that might hinder the adoption or scalability of your framework in practical settings? It would be valuable to understand the feasibility and potential constraints of implementing BirDRec in real-world scenarios.

**Limitations:**

The authors missed an opportunity to compare their approach with other recent methods that address the problem of unreliable data in SRSs. By including such a comparison, the authors could provide a more comprehensive evaluation of their proposed framework and highlight its advantages or limitations in relation to existing solutions. This addition would enhance the paper's contribution to the field.

---

> ### Author Rebuttal · Authors · 2023-08-09
>
>
> Thanks for your reviews.
>
> > Weakness 1
>
> We exactly follow the scientific research standards and common practice in SRSs, that is, selecting (around 3~6) representative benchmark datasets that are widely adopted by SOTAs [1, 2, 4] for evaluation. We have tried our best to cover various domains ranging from movie (ML-1M), video (QK-Video), to E-commerce (Amazon-Beauty) and location-based social networks (Yelp).
>
> Due to space limitation, we believe no one and no method can enumerate all datasets for evaluation. Thus the generalizability of SOTAs cannot be fully guaranteed on datasets that have not been tested.
>
> > Weakness 2 & Question 3
>
> Our solution is indifferent to the cause of unreliable instances. Regardless of the reasons causing an unreliable instance, it will involve either unreliable input or unreliable target, or both. All of these can be handled by BirDRec.
>
> We acknowledge that analyzing the causes of unreliable data might help devise solutions. However, investigating why a user interacts with irrelevant items is a challenging psychological problem [3], which is beyond the scope of this paper. We may leave it as our future work.
>
> > Weakness 3
>
> The time and space complexity of our proposed framework are respectively analyzed in Sections 4.1 and 4.2.
>
> The main limitations of BirDRec are twofold, which will be highlighted in our camera-ready.
>
> - **Limited improvements on storage cost for smaller item sets:** As mentioned in lines 293-295, we acknowledge that the storage cost of BirDRec is marginally higher than STEAM's on ML-1M. This is primarily because the rectification sampling strategy with Self-ensemble has a less apparent advantage on datasets with smaller item sets (see lines 246-249). To be specific, the storage cost reduction for calculating weighted average prediction scores is from $O(|V|*H)$ to $O(K)$ for each instance. Thus if $|V|$ is small, the benefit of this reduction will be less obvious.
> - **Marginally higher computational cost than backbone:** Although BirDRec is significantly faster than the latest robust SRS (STEAM), it is worth noting that BirDRec is 1.6 times on average slower than its backbone model SASRec (as depicted in Fig 3) in each training epoch. This increased training time could be a practical concern in systems with **extremely large-scale** datasets and real-time recommendation demands.
>
> > Question 1
>
> The details and proofs of all the lemmas and theorems can be found in Appendix.
>
> > Question 2
>
> **For datasets**, we exactly follow the scientific research standards and common practice in SRSs, that is, selecting (around 3~6) representative benchmark datasets that are widely adopted by SOTAs [1, 2, 4, 5] for evaluation. Typically, they are selected by considering the following criteria: domains, sizes, sparsity levels, and average sequence lengths as shown in Tab 1.
>
> **For metrics**, these evaluation metrics are selected by following SOTAs in SRSs [1, 2, 4]. Higher metric values indicate better ranking performance. We can add the following detailed explanations in Appendix.
>
> Suppose there are $M$ users, and $R_{u}$ is the full recommendation list (RL) for user $u$.  Let $R_{u}[j]$ be the j-th item in $R_{u}$, and $R_{u}[1:N]$ be the Top-$N$ RL. $i_{u}^{t}$ is $u$'s intereacted item in the test set. $I(x)$ is an indicator function whose value is $1$ when $x > 0$, and $0$ otherwise. The formal definitions of HR, NDCG, and MRR are as follows.
>
> **HR (Hit Ratio)** gives the percentage of users that can receive at least one correct recommendation from the Top-N RL:
>
> $$HR@N=\frac{1}{M}\sum_{u=1}^{M}I(|R_{u}[1:N]\cap\{i_{u}^{t}\}|).$$
>
> **NDCG (Normalized Discounted Cumulative Gain)** evaluates the ranking performance by measuring the positions of correct recommended items:
>
> $$NDCG@N=\frac{1}{M}\sum_{u=1}^{M}\frac{1}{Z}DCG@N=\frac{1}{M}\sum_{u=1}^{M}\frac{1}{Z}\sum_{j=1}^{N}\frac{2^{I(|\{R_{u}[j]\}\cap \{i_{u}^{t}\}|)}-1}{\log_2(j+1)},$$
>
> where Z, as a normalization constant, is the maximum possible value of DCG@N.
>
> **MRR (Mean Reciprocal Rank)** also evaluates the ranking performance via the rank position of the correct recommended item $i_{u}^{t}$ (denoted by $rank_{i_{u}^{t}}$) in the RL:
>
> $$MRR=\frac{1}{M}\sum_{u=1}^{M}\frac{1}{rank_{i_{u}^{t}}}.$$
>
> > Question 4
>
> BirDRec would be particularly beneficial for domains where users' behaviors can be occasionally influenced by external distractions. For example, in movie domain, users may receive recommendations from friends once in a while. Meanwhile, it also benefits from datasets with large item sets, leading to significant improvements in storage cost compared with the latest robust SRS STEAM [2].
>
> > Limitation 1
>
> We did compare our model with 3 recent SOTA robust SRSs, i.e., BERD [1] (IJCAI-21), FMLP-Rec [4] (TheWebConf-22), and STEAM [2] (TheWebConf-23). They aim to tackle unreliable targets, unreliable input, and both, respectively. The results are in Tab 3.
>
>
> References
>
> [1] Sun et al. Enhancing sequential recommendation by eliminating unreliable data. In IJCAI 2021.
>
> [2] Lin et al. A self-correcting sequential recommender. In TheWebConf 2023.
>
> [3] Cosley et al. Is seeing believing? How recommender system interfaces affect users’ opinions. In SIGCHI 2003.
>
> [4] Zhou et al. Filter-enhanced mlp is all you need for sequential recommendation. In TheWebConf 2022.
>
> [5] Yuan et al. Tenrec: A Large-scale Multipurpose Benchmark Dataset for Recommender Systems. In NeurIPS 2022.

---

> > ### Comment · Reviewer_1jEu · 2023-08-17
> >
> > I recommend that the author incorporate this section into the main body of the revised paper. While I am inclined to give a higher score based on this addition, I believe it's crucial to consider the feedback from other reviewers as well. I have no further questions.

---

> > > ### Author Response · Authors · 2023-08-17
> > >
> > > Thank you for taking the time to review our response. We sincerely appreciate your willingness to award a higher score. Your feedback is invaluable, and we will thoughtfully address your comments and integrate the mentioned section into the main body of our camera-ready submission.

---

### Official Review · Reviewer_E67v · 2023-07-07

**Soundness:** 3 good
**Presentation:** 3 good
**Contribution:** 2 fair
**Rating:** 5
**Confidence:** 3

**Summary:**

The authors proposed Bidirectional Data Rectification (BirDRec) framework that can be incorporated into existing Sequential recommender systems (SRSs) to tackle unreliable data. In addition, the authors also adopt rectification sampling strategy.

The authors also provided theoretical guaranteed for BirDRec and conducted various experiments on the robustness of the framework.


**Strengths:**

Strengths:

-	The idea is straightforward and easy to follow
-	The authors provided theoretical guaranteed for BirDRec and conducted various experiments on the robustness of the framework.
-	Theoretical aspects are well-written
-	The authors also provided source code for verification.


**Weaknesses:**

Weaknesses:

-	Some of the experimental results are not convincing such as the performance of baselines in Table 3. The authors need to provide more explanation and clarification such as why BERD is good for Movielens1M but STEAM is better for Yelp.
-	It’s not clear which parts of the architecture in Figure 2 made the performance increases dramatically
-	Comparing BirDRec and SASRec for hyper-parameters analysis in Appendix C.2 and C.3 seems a bit ‘unfair’. In my opinion, we should compare with most recent baselines, so at least we can compare with Figure 5 in the main paper.
-	How much time did it take to do grid search across models in Table 1 from Appendix Page 13?


**Questions:**

Please refer to my comments above

---

> ### Author Rebuttal · Authors · 2023-08-09
>
>
> Thank you for your invaluable comments.
>
> ## Weaknesses
>
> > W1
>
> The possible reasons are twofold: the item-wise correction process of STEAM, and the longer interaction sequences on ML-1M (see Tab 1).
>
> To correct each instance, STEAM selects one type of operation from ‘keep’, ‘delete’, and ‘insert’ for **each item** in the instance. Such correction is more likely to make mistakes on long sequences as one single wrong operation will render the entire sequence unreliable for training. By contrast, BERD ONLY cares whether the **target item** is matched with the input items given an instance, so its performance is insensitive to the length of the sequence.
>
> Besides, the mistakes made by STEAM can be more serious than BERD, as STEAM can modify reliable instances into unreliable ones by deleting relevant items or inserting irrelevant items. In contrast, BERD simply abandons some instances that might be unreliable, but would not generate new unreliable instances. Such negative impacts will be amplified on datasets with long sequences where STEAM is more inclined to make mistakes.
>
>
> > W2
>
> Based on our ablation study (see lines 296-305 and Fig 4), the performance gain of separately adding DRUT, DDUI, and Self-ensemble is 45.7%, 43.3%, and 3.1%, respectively. Thus the ranking of their importance should be DRUT, DDUI, and Self-ensemble. We will complement the ablation study by adding this in the camera-ready.
>
>
> > W3
>
> All the figures in C.2 and C.3 show the performance comparison with SASRec itself and our BirDRec with SASRec as backbone. These figures mainly aim to investigate the effects of different hyper-parameters on BirDRec, so as to help determine the best setting for the hyper-parameters. Hence, the comparison is fair.
>
> For a more comprehensive visualization, we have added the three recent baselines (BERD: IJCAI-21, FMLP-Rec: TheWebConf-22, STEAM: TheWebConf-23) into C.2 and C.3. Note that their performance is already available in Tab 3, and their per-epoch accuracy is available in the training logs. The results show that BirDRec still performs the best.
>
> > W4
>
> It takes around three months to complete the grid search in Tab 1 in Appendix. This aims to help find out optimal hyper-parameter settings for all methods, thus delivering a fair and rigorous comparison. To improve efficiency, we accelerated this process with two strategies following SOTAs [1, 2].
>
> **First**, we adopt the early stop mechanism [1] to terminate the model training if the performance does not increase in 20 consecutive epochs. **Second**, we split the hyper-parameters of each method into three independent groups, including learning rate-related ones (e.g. learning rate, batch size), model capacity-related ones (e.g. embedding size, number of attention heads), and the others. Then we conduct the grid search on each group separately [2], and the hyper-parameters not in this group are set as suggested in their original papers. Ultimately, the optimal setting of each group together forms the final optimal settings of a specific method.
>
>
> References
>
> [1] Goodfellow et al. Deep Learning. 2016.
>
> [2] Yoshua Bengio. Practical Recommendations for Gradient-Based Training of Deep Architectures. Neural Networks: Tricks of the Trade. 2012.

---

> > ### Comment · Reviewer_E67v · 2023-08-20
> > **Thanks for rebuttal**
> >
> > Thanks for the rebuttal. I would suggest the authors to add W2 and W3 to the main body. I decided to keep my score after taking a closer look at other reviews and responses.

---

### Official Review · Reviewer_rz4h · 2023-07-09

**Soundness:** 3 good
**Presentation:** 3 good
**Contribution:** 3 good
**Rating:** 6
**Confidence:** 4

**Summary:**

This paper focuses on the inconsistency between user’s true preference with interaction history. Specifically, the authors propose a theoretically guaranteed data rectification strategies based on SRS predictions to tackle both unreliable input and targets for more robust SRS. Besides, the authors also devise a rectification sampling strategy and adopt a self-ensemble mechanism to ensure better scalability. Experiments on four real-world datasets show the superiority of the proposed method.

**Strengths:**

1)	The research problem of rectification of interaction data is very interesting.

2)	Theorem analysis for the proposed modules.

3)	The proposed model can be compatible with most existing models.

4)	Extensive experiments with respect to effectiveness and efficiency are conducted.

**Weaknesses:**

1)	Some details are not clear, for example, what’s the formulation of the Self-ensemble Mechanism?

2)	The time comparison is important is important for real application. However, Figure 3 compares SASRec, STEAM, and BirDRec simultaneously, which makes the increasing of time comparing SASRec and BirDRec not clear. More analysis is necessary.

3)	The experimental datasets are all small datasets with less than 100K users and items. Besides, the improvements are too large which is not very common.

4)	No online results are given. Too much hyper-parameters, which limit the applicability of the model.

5)	Many parameters are pre-defined, without careful tuning.

**Questions:**

1)	Theorem 4 indicates that the relative rank of an item over a list of randomly sampled items can approximate this item’s relative rank over the full item set. However, if K is too smaller than |V|, the randomly sampled items may still introduce so many noises?

2)	The space complexity reduction of Self-ensemble Mechanism if from K*H to K, but not from |V|*H to K?

**Limitations:**

1)	The proposed model seems to be too complicated to be deployed in real industrial environments.

2)	Besides sequential recommendation, can this model employed in other recommendation tasks, for example, rating prediction, CTR prediction?

---

> ### Author Rebuttal · Authors · 2023-08-09
>
> Thanks for your invaluable reviews.
>
> ## Weaknesses
>
> > W1
>
> We did have the formulation of the Self-ensemble mechanism (see lines 238-245).
>
> > W2
>
> Thanks for bringing up the clarity issue with Fig 3(a). The table below shows the per-epoch time cost (seconds) of SASRec and BirDRec:
>
> | | ML-1M | Beauty | Yelp | QK-Video
> ---|---|---|---|---
> SASRec | 14.1  | 5.0    | 5.7  | 44
> BirDRec | 28.5  | 7.9    | 7.5  | 74
>
> We will clarify this by adding values for each bar in Fig 3(a) in the camera-ready.
>
> > W3
>
> The dataset selection exactly follows the common practice in SRSs. They are representative benchmark datasets and have been widely adopted by SOTAs [1, 2, 3] in SRSs. Although the numbers of users and items are less than 100K, the total interactions are in a large scale, e.g., QK-Video has larger than 2M interactions.
>
> **BirDRec vs. non-robust SRSs.** It is common for robust SRSs to achieve large improvements over non-robust SRSs due to the negative effects of unreliable data in misleading SRSs. The table below shows the average improvements of BirDRec and the best robust SRS over six non-robust SRSs. Both methods achieve significant improvements.
>
> | |   FPMC  |  Caser  | GRU4Rec | SASRec | BERT4Rec |   MAGNN
> ---|---|---|---|---|---|---
> Best Robust SRS |   31.13%  |32.72%  |42.52% | 17.73%  | 21.81%  | 10.45%
> BirDRec | 38.91% | 40.65% | 64.81% | 49.27% | 51.06% | 25.94%
>
> **BirDRec vs. Robust SRSs (see Tab 3).** The relative improvement achieved by the best SOTA robust SRS against other robust SRSs is 23.45% on HR. BirDRec wins the best SOTA robust SRS by a lift of 25.68%. The reasons are threefold. BirDRec not only detects but also rectifies unreliable data; BirDRec handles both unreliable input and targets; and BirDRec is error-bounded.
>
>
> > W4
>
> Online evaluation for SRSs is challenging due to privacy and ethical concerns w.r.t. accessing real-world platforms and user interaction data. We follow SRS SOTAs [1, 2, 3] to report offline results on public datasets.
>
> Beyond shared hyper-parameters (e.g., embedding size, batch size), BirDRec only has **4 additional** hyper-parameters (see Fig 5). SOTA robust SRSs [1, 2] normally have **5 additional** hyper-parameters.
>
> > W5
>
> We have carefully tuned hyper-parameters of each method with the grid search (see Fig 5 in the main paper and Tab 1 in Appendix). To improve efficiency, we accelerated this process with two strategies following SOTAs [4, 5]. **First**, we adopt the early stop mechanism [4] to terminate the model training if the performance does not increase in 20 consecutive epochs. **Second**, we split the hyper-parameters of each method into 3 independent groups: learning rate-related ones, model capacity-related ones, and the others. Then we conduct the grid search on each group separately. The optimal setting of each group forms the final optimal setting [5].
>
> ## Questions
>
> > Q1
>
> Theoretically, based on Theorem 4, larger $K$ yields a more accurate estimation of the true rank. If $K$ is too smaller than $|V|$, the upper bound of Theorem 4 might be high, thus introducing noise. Empirically, we found that $K\ge 10$ is sufficient to reach promising accuracy on all datasets. The ratio of $K$ to $|V|$ is from 1/300 to 1/4000 across different datasets. The possible reasons are twofold.
> - Each rectification pool is initialized properly with an instance's succeeding or preceding items, which are potentially better substitutions.
> - Each rectification pool is progressively improved during training as it is continuously updated by adding newly sampled high-scored items.
>
> > Q2
>
> The reduction is from $|V|*H$ to $K$ (see lines 234-251). **Without Self-ensemble**, each instance has to maintain prediction scores with all $|V|$ items in all $H$ previous epochs to trace back historical scores and calculate the weighted average scores. As we cannot forecast which item will be added to the rectification pool. **With Self-ensemble**, the weighted average scores of $K$ items in a rectification pool can be directly approximated by a self-ensembled model.
>
> ## Limitations
>
> > L1
>
> Based on Fig 3, BirDRec is much more efficient than the latest SOTA Robust SRS STEAM, and only 1.6 times slower than its backbone. Besides, BirDRec is particularly beneficial for datasets with large item sets (see lines 246-249) thus being applicable for real environments.
>
>
> > L2
>
> BirDRec can be adapted to other recommendation tasks, as long as timestamps of user-item interactions are available. Thus, the problem can be first formulated as a sequential recommendation task. Then, the expected output can be generated by properly adding extra neural layers.
>
> For rating prediction, the output of sequential recommendation (i.e., prediction scores for all candidate items) could be first mapped into $[0, 1]$ by a softmax layer, and then re-scaled based on the rating range. For CTR prediction, each CTR can be computed by a softmax layer over the output of sequential recommendation.
>
> References
>
> [1] Sun et al. Enhancing sequential recommendation by eliminating unreliable data. In IJCAI 2021.
>
> [2] Lin et al. A self-correcting sequential recommender. In TheWebConf 2023.
>
> [3] Qiu et al. Contrastive Learning for Representation Degeneration Problem in Sequential Recommendation. In WSDM 2022.
>
> [4] Goodfellow et al. Deep Learning. 2016.
>
> [5] Yoshua Bengio. Practical Recommendations for Gradient-Based Training of Deep Architectures. Neural Networks: Tricks of the Trade. 2012.

---

> > ### Comment · Reviewer_rz4h · 2023-08-20
> > **Reply to Authors' Rebuttal**
> >
> > Thanks for your rebuttal. Your responses have addressed most of my concerns.

---

### Decision · Program_Chairs · 2023-09-21

**Decision:**

Accept (poster)

**Comment:**

All five reviewers unanimously recommend acceptance (ranging from borderline accept to strong accept).  Reviewers request that the explanations and clarifications from the rebuttal should be integrated into the final paper version (or supplementary material).